# Genomic alterations and evolution of cell clusters in metastatic invasive micropapillary carcinoma of the breast

Qianqian Shi [1,10], Kang Shao [2,3,10], Hongqin Jia[1,4,10], Boyang Cao[2,3,10], Weidong Li[1,4,5,6,7], Shichen Dong[2,3], Jian Liu[1,4,5,6,7], Kailiang Wu[1,4,5,6,7], Meng Liu[2,3], Fangfang Liu[1,4,5,6,7], Hanlin Zhou [2,3], Jianke Lv[1,4,5,6,7], Feng Gu [1,4,5,6,7], Luyuan Li[8], Shida Zhu[2,3], Shuai Li [1,4,5,6,7,11✉], Guibo Li [2,3,9,11✉] & Li Fu [1,4,5,6,7,11✉]

Invasive micropapillary carcinoma (IMPC) has very high rates of lymphovascular invasion and lymph node metastasis and has been reported in several organs. However, the genomic mechanisms underlying its metastasis are unclear. Here, we perform whole-genome sequencing of tumor cell clusters from primary IMPC and paired axillary lymph node metastases. Cell clusters in multiple lymph node foci arise from a single subclone of the primary tumor. We find evidence that the monoclonal metastatic ancestor in primary IMPC shares high frequency copy-number loss of *PRDM16* and *IGSF9* and the copy number gain of *ALDH2*. Immunohistochemistry analysis further shows that low expression of IGSF9 and PRDM16 and high expression of ALDH2 are associated with lymph node metastasis and poor survival of patients with IMPC. We expect these genomic and evolutionary profiles to contribute to the accurate diagnosis of IMPC.

[1] Department of Breast Cancer Pathology and Research Laboratory, Tianjin Medical University Cancer Institute and Hospital, 300060 Tianjin, China. [2] BGI-Shenzhen, 518120 Shenzhen, China. [3] BGI College & Henan Institute of Medical and Pharmaceutical Sciences, Zhengzhou University, 450000 Zhengzhou, China. [4] National Clinical Research Center for Cancer, 300060 Tianjin, China. [5] Key Laboratory of Cancer Prevention and Therapy, 300060 Tianjin, China. [6] Tianjin's Clinical Research Center for Cancer, 300060 Tianjin, China. [7] Key Laboratory of Breast Cancer Prevention and Therapy, Tianjin Medical University, Ministry of Education, 300060 Tianjin, China. [8] State Key Laboratory of Medicinal Chemical Biology and College of Pharmacy, Nankai University, 300071 Tianjin, China. [9] Shenzhen Key Laboratory of Single-Cell Omics, 518120 Shenzhen, China. [10] These authors contributed equally: Qianqian Shi, Kang Shao, Hongqin Jia, Boyang Cao. [11] These authors jointly supervised this work: Shuai Li, Guibo Li, Li Fu. ✉email: shuaili@tmu.edu.cn; liguibo@genomics.cn; fuli@tmu.edu.cn

Invasive micropapillary carcinoma (IMPC) was first described in breast cancer by Fisher et al. in 1980[1] and was officially named IMPC in 1993[2]. It was also first listed as a special histological subtype of breast cancer in the 2003 edition of the World Health Organization (WHO) Classification of Tumors of the Breast[3]. To date, IMPC has been reported in several organs, including the bladder[4], colon[5], lung[6], pancreas[7], etc. IMPC in lung cancer was also listed as a special histological type in the 2015 edition of the WHO Classification of Lung Tumors[8]. Morphologically, IMPC cells are adhered to micropapillary or morula-like structures without fibrovascular cores and are located within empty stromal spaces[2]. Tumor cell clusters display an inverted growth pattern, in which the luminal aspect of the cell appears on the outer surface of the cell cluster[9]. Clinically, compared with invasive ductal carcinoma (IDC), IMPC exhibits a higher rate of lymph vascular invasion and lymph node metastasis and a poorer prognosis[10–13]. The unique clustered growth pattern and aggressive biological behaviors make IMPC a good model for studying cancer invasion and metastasis.

Fu et al. further noted that micropapillary and/or glandular tumor cell clusters in tumor tissues from patients with breast cancer are accompanied by lymphatic invasion and lymph node metastasis, even if the tumors are small and/or the tumor tissue contains few IMPC components[10]. Furthermore, Fu et al. performed a prospective clinicopathological study of IMPC in the breast in 1994. Based on the results, the authors proposed the pathological diagnostic criteria to "make a diagnosis as long as there are IMPC components and mark the proportion"[14]. In addition, Fu et al. found that IMPC tumor cells are arranged in micropapillary and/or glandular tumor cell clusters not only in primary tumors but also in lymphovascular and lymph node metastatic foci[14]. Subsequently, a series of studies by Fu et al. successfully examined the pathology, clinical features, and metastatic mechanism of the "clustered metastasis of IMPC tumor cells"[15–22]. In this study, we aimed to assess the genomic features of IMPC and identify the metastasis-associated genes that govern the initiation and growth of IMPC metastasis. The findings may help effectively prevent and treat IMPC.

In the present study, we performed whole-exome sequencing (WES) and whole-genome sequencing (WGS) on freshly-frozen primary IMPC and paired normal tissue, and characterize the mutational and copy-number variation (CNV) spectra of the IMPC genome. We then conduct topographic cell cluster sequencing (TCCS) of individual cell clusters extracted from formalin-fixed paraffin-embedded (FFPE) specimens of the primary IMPC and matched lymph node metastatic lesions to characterize the CNV and genomic features of metastatic IMPC. Thus, in this study, we perform the genomic analysis of metastatic versus primary IMPC and reveal the genomic basis of CNV evolutionary routes in IMPC tumor cell clusters.

## Results

**An overview of the somatic genetic alterations in IMPC.** We performed WES of 17 pairs of freshly frozen IMPC tumor and normal tissues to establish an overview of the genomic mutation spectrum of IMPC (the clinicopathological and sequencing information is listed in Supplementary Data 1, and the experimental flow chart is depicted in Supplementary Fig. 1), reaching an average sequencing depth (i.e., the number of times each base had been sequenced) of more than 155× per sample (median = 235×, standard error of the mean [SEM] = 37.23). We identified 3569 somatic mutations in the primary IMPC. After the exclusion of 2231 silent mutations, 1338 mutations remained, including 1190 missense, 80 nonsense, 53 splice-site, 14 indels, and 1 nonstop mutation (Supplementary Fig. 2A and Supplementary

Data 2, non-silent somatic mutations in IMPC). The variant type was dominated by a single-nucleotide variant (SNV), and the most frequent mutation event was a C-to-T transition (Supplementary Fig. 2A), which is a common event in breast cancer (Supplementary Fig. 2B) and other epithelial tumors[23]. The mean IMPC tumor mutation burden (TMB) was 2.32, which was slightly higher than the mean TMBs of The Cancer Genome Atlas (TCGA) breast cancer (mean TMB = 1.75) and TCGA subtype PAM50 luminal B breast cancer datasets (mean TMB = 1.62), but the differences were not significant (Student's $t$ test, $P = 0.178$ and $P = 0.108$, respectively, Supplementary Data 2). Additionally, the most frequently mutated genes in IMPC were TP53 (24%, 4/17), followed by CDC27, CELSR2, CEP192, KIF26A, and RYR2 (18%, 3/17, Fig. 1a), while the other mutated genes were present in <3 samples. Since the number of mutations in the 17 samples might be limited to such an extent that the frequency of gene mutations might not reflect the true genetic alteration spectrum of IMPC, we also examined the genes that were reported to have a high mutation frequency in breast cancer[24], such as PIK3CA, AKT1, MAP3K1, FOXA1, GATA3, NCOR1, PTEN, PTPRD, and RUNX1 (Fig. 1a). However, the mutation frequency of PIK3CA, which is commonly mutated in breast cancer, in IMPC was only 12% (2/17). The mutation frequencies of other commonly mutated genes, such as AKT1, MAP3K1, FOXA1, and GATA3, were only 6% (1/17) (Fig. 1a). Interestingly, PIK3CA is the most frequently mutated gene in the TCGA breast cancer cohort (32%) (Fig. 1b), and its mutation frequency in the luminal B subtype of breast cancer is 29% (Fig. 1c). The pathway analysis of all non-silent mutations in somatic genes showed that the RTK-RAS pathway involved the most samples (52.94%, 9/17) (Supplementary Fig. 2C), and the samples with more lymph node metastases were mostly enriched in this pathway; thus, the RTK-RAS pathway may be an important pathway (Supplementary Fig. 2D). The analysis of 96 signatures based on the three-nucleotide context revealed that C>T mutations accounted for the majority of the mutations, consistent with TCGA breast cancer database (Supplementary Fig. 3). Furthermore, a deconstructSigs analysis revealed that mutational signatures of APOBEC (signatures 2 and 13), age (signature 1), homologous recombination deficiency (HRD) (signature 3), and DNA mismatch repair signatures (signatures 6 and 15) were relatively more frequently observed in our IMPC samples (Supplementary Fig. 4 and Supplementary Data 3).

**Genomic CNVs correlates with lymph node metastases of IMPC.** Subsequently, we performed low-depth WGS and analyzed the CNV in 17 frozen IMPC samples (Fig. 2a, b). In addition to observing the CNV features reported in the literature in the IMPCs[25–27], such as the most frequently reported arm changes 1q, 8q, and 20q gains and 16q and 19q loss (the frequency of these arms are shown in Supplementary Data 4 arm-level comparison), the CNV profile from GISTIC2.0 also showed a novel focus of the IMPC-specific genomic gain/amplification regions in chromosome 10q22.3 (a gain was detected in 3 samples [17.65%, 3/17], and amplification was detected in 3 samples [17.65%, 3/17]) (Fig. 2d). This region contains the cancer-related gene NUTM2B, which is associated with kidney clear cell sarcoma[28] and ovarian endometrioid stromal sarcoma[29]. Additionally, the CNV distribution in the 17 IMPC samples was clustered into CNV low and CNV high groups according to the CNV burden (Student's $t$ test, $P = 0.0034$) (Fig. 2a, e). The CNV low group ($n = 9$, P11, P17, P36, P43, P42, P22, P38, P44, and P39) exhibited relatively low frequencies of CNVs. In sharp contrast, the CNV high group ($n = 8$, P45, P8, P19, P41, P37, P40, P35, and P14) showed more widely distributed CNVs

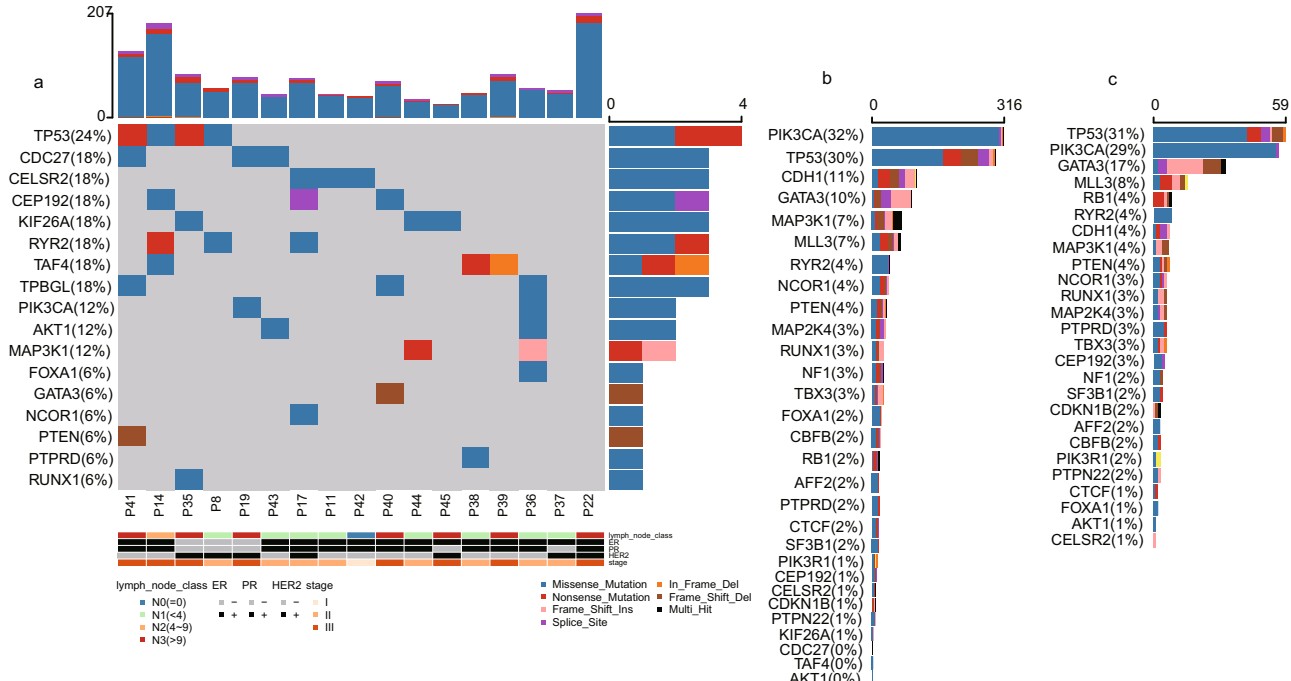

**Fig. 1 IMPC genomic somatic mutation landscape in 17 freshly frozen IMPC tumor-normal tissue pairs. a** Non-silent somatic mutation patterns and frequencies of the most highly mutated genes (present in at least 3 samples: *TP53, CDC27, CELSR2, CEP192, KIF26A, RYR2, TAF4, TPBGL*, and known breast cancer-related genes *PIK3CA, AKT1, MAP3K1, FOXA1, GATA3, NCOR1, PTEN, PTPRD*, and *RUNX1*; known breast cancer-related genes *MAP2K4, MLL3, CDH1, PIK3R1, CBFB, TBX3, CTCF, SF3B1, CDKN1B, RB1, AFF2, NF1*, and *PTPN22* that are not shown in the figure did not contain mutations). The percentages on the left show the frequencies of gene mutation, and the bars on the top and right represent the number of different mutations in each sample and the sample number in each gene, respectively. **b** The same non-silent somatic gene mutations as shown in **a** with patterns and frequencies for TCGA breast cancer dataset. **c** The same non-silent somatic gene mutations as shown in **a** with patterns and frequencies for TCGA luminal B breast cancer dataset.

characterized by gains and losses (Fig. 2c). Interestingly, on average, the CNV high group had a substantially higher lymph node stage than the CNV low group (Supplementary Table 1). This finding indicated that the frequency of CNVs was associated with lymph node metastasis (CNV high versus CNV low groups, Mann–Whitney $U$-test, $P = 0.045$).

Based on the CNV data, several special CNV regions were identified, including gain peaks containing known breast cancer driver genes such as *ERBB2* and *NOTCH2*, as well as the loss peaks containing some known tumor suppressor genes such as *PRDM16* and *STK11* (Fig. 2d and Supplementary Data 4, CNV focal peak gene annotation). Furthermore, we used the *t* test to identify significantly different focal regions (Student's *t* test, $P < 0.05$) between the CNV high and CNV low groups. We found some loss regions (Supplementary Data 4, CNV focal peak gene annotation) that contain a set of cancer-related genes, such as *PCM1*, *PRDM16*, and *STK11*. These genes displayed a copy-number loss in the CNV high group and may play a key role in the development of lymph node metastasis of IMPC.

As we obtained both SNV and CNV data from 17 freshly frozen IMPC samples, we investigated which type of mutation was associated with the high lymph node metastatic potential of IMPC. We compared CNVs with SNVs to determine which type of variant was more strongly associated with the high lymph node metastatic potential of IMPC. Here the Pyclone method[30] was used to analyze heterogeneity and determine the association between the lymph node metastasis and genomic variations (including CNVs and SNVs). To analyze the heterogeneity of the SNV data, a gradient threshold was set to determine the number of characteristic gene mutations (N). The correlation between mutational heterogeneity and the potential of lymph node metastasis peaked at $N = 3$ and then gradually decreased,

suggesting that an increase in the value of N would be meaningless. We therefore set the gradient threshold to 1–8 (Fig. 2f). Moreover, we calculated the correlation coefficients of lymphatic metastatic number and SNVs and those of lymphatic metastatic number and CNVs, respectively (Fig. 2f). The highest correlation coefficient between lymph node metastasis and SNVs was 0.21 when using a subclonal population threshold of $N = 3$, whereas the correlation coefficient was 0.44 (0.44 > 0.21) between the lymph node metastatic potential and CNV levels. Based on this finding, lymph node metastasis was more significantly associated with CNVs than SNVs. Furthermore, we used the Jaccard similarity index (JSI)[31] to quantify mutational similarities reflected by the tumor metastatic potential at the SNV and CNV levels. The JSI values of CNVs were significantly higher than those of SNVs (Wilcoxon rank-sum and signed-rank tests, $P = 0.0001$) (Supplementary Fig. 5A). Additionally, we used the method reported by Ciriello et al.[32] to calculate the difference between SNV and CNV burdens in our IMPC dataset and found that the CNV burden was higher than the SNV burden, indicating that CNVs were significantly predominant in IMPC (Supplementary Fig. 5B). Thus, the potential of lymph node metastasis is more significantly associated with CNVs than with SNVs.

**CNV in pure IMPC and mixed IMPC-IDC samples using multiple-cell cluster sequencing.** Subsequently, we isolated IMPC multiple-cell clusters from the FFPE IMPC specimens from 8 pure IMPC and 21 mixed IMPC-IDC cases using the laser-capture microdissection (LCM) method. Then the genomic DNA was subjected to low-coverage WGS to determine genome-wide CNVs (the experimental flow chart is shown in Supplementary Fig. 1D; the clinicopathological and sequencing data are

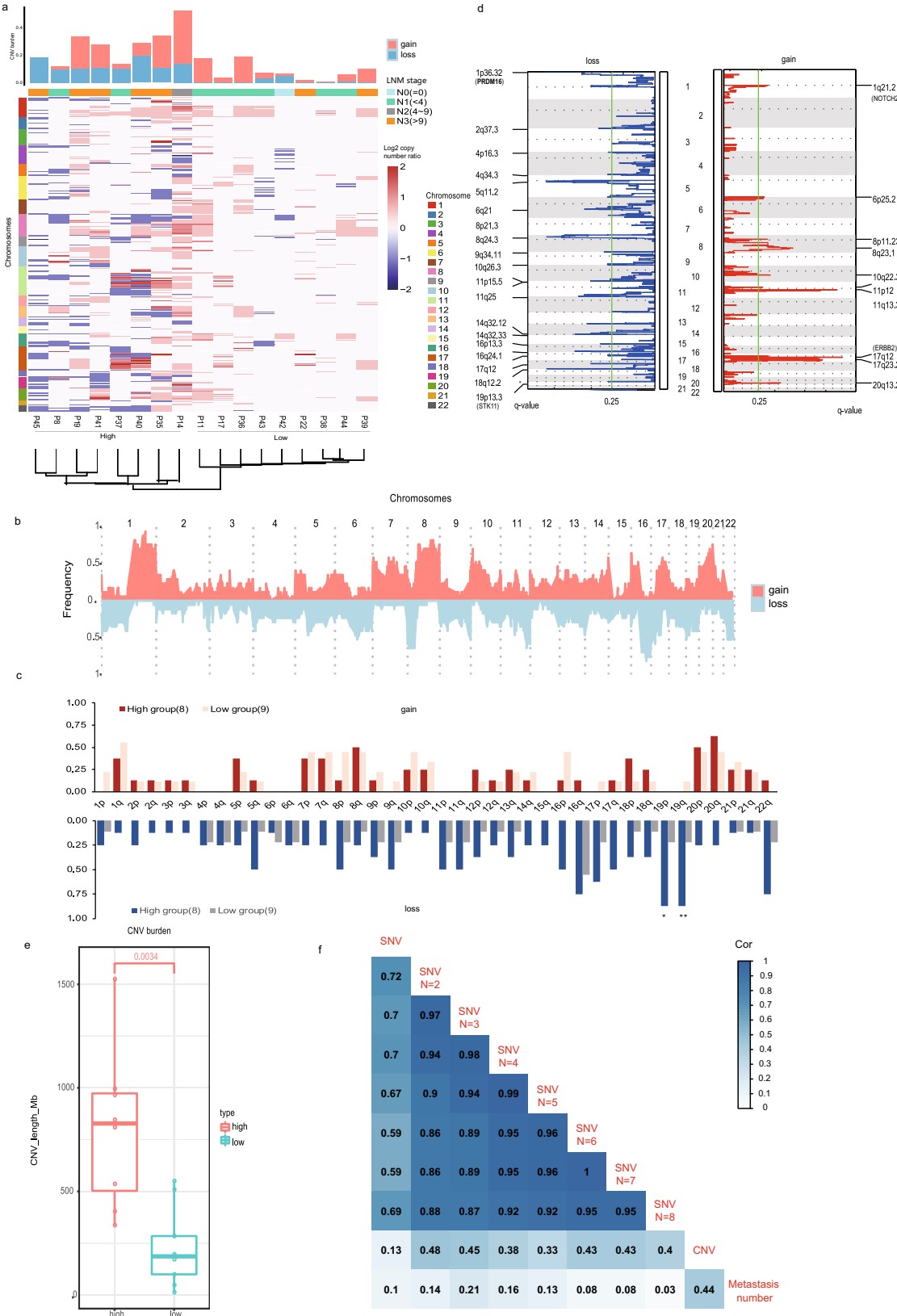

shown in Supplementary Data 5). Pure IMPC and IMPC components in mixed IMPC-IDC shared common CNVs in 42 genes (those observed in both the pure IMPC and IMPC component in mixed IMPC-IDC), as well as 48 and 90 CNVs in genes that were specifically altered in pure IMPC and the IMPC component in mixed IMPC-IDC, respectively (Fig. 3b and Supplementary

Data 6). This finding suggested that the IMPC component in mixed IMPC-IDC contained more specific CNVs than pure IMPC, but CNV burden has no significant difference with pure IMPC (Supplementary Fig. 10a; Student's *t* test, *P* = 0.097). Moreover, some high-frequency CNVs were identified in genes in each category, such as the gain of *CDK6* and loss of *ZFHX3* and

**Fig. 2 IMPC genomic CNVs in 17 freshly frozen IMPC tumor-normal tissue pairs. a** Copy-number-based hierarchical clustering based on the Control-FREEC region results. Heatmap with log2 copy-number ratios across the genome; 0 indicates no change, a positive value indicates a gain, and a negative indicates a loss (all values range from −2 to 2). The CNV burden is shown on the top of the heatmap. Patient ID with IMPC and lymph node metastasis (LNM) stages are presented on the right, and the chromosomal locus is presented on the left. Two groups are clustered: one contains more CNVs and is called the CNV high group, while the other contains fewer CNVs and is called the CNV low group. **b** Frequency of copy-number aberrations across the 17 bulk samples. Chromosomes 1–22 are positioned along the x-axis, while the y-axis shows the frequency of copy-number gains (salmon) and losses (light blue). **c** Arm-level frequency of copy-number aberrations across the 17 bulk samples determined using the GISTIC2.0 method. A significant loss in chromosome 19p and 19q was observed in all 17 samples and between the two CNV groups (Supplementary Data 4). A single asterisk (*) represents both P value (joint hypotheses test) and GISTIC q value (multiple hypothesis testing using the BH FDR method) < 0.05, while double asterisks (**) represents P value < 0.05 and GISTIC q value < 0.01; the exact value is present in sheet 1 of Supplementary Data 4. **d** Identification of focal recurrent copy-number losses (left panel) and gains (right panel) using the GISTIC2.0 method. **e** Box plots of the CNV burden (with the box plot center, box, whiskers, and points corresponding to the median, interquartile range, 1.5× interquartile range, and outliers, respectively). Student's t test for total CNV length burden between the CNV high and CNV low groups; the two groups were significantly different (P = 0.0034). **f** Relationship between heterogeneity and the lymph node metastasis number in patients with IMPC. Heterogeneity was evaluated using Pyclone with SNVs and CNVs. A gradient threshold was set for SNVs (N = 1–8). Each number represents the correlation of its corresponding horizontal and vertical coordinate conditions, including heterogeneity among SNVs, CNVs, and the lymph node metastasis number. The metastatic potential was evaluated by lymph node metastasis number; a higher lymph node metastasis number indicates the involvement of more lymph nodes and the higher metastatic potential of IMPC.

POLE in pure IMPC and the gain of PTPRC and ATK3 and loss of BCL9L and DICER1 in mixed IMPC-IDC (recurrently altered genes were present in at least half of samples; the frequency of these genes is shown in Supplementary Data 6).

We next isolated IMPC and IDC cell clusters from primary mixed IMPC-IDC specimens and obtained sequencing data from 14 pairs to determine whether the different pathological components in mixed IMPC-IDC have distinct genomic variations (Supplementary Data 5). A comparison of these pairs showed that CNV gain/loss was more frequently observed in the IMPC component than in the IDC component (Student's t test, P = 0.003; Fig. 3a). This observation differs from a published finding that both components of primary IMPC harbor strikingly similar genomic profiles[33]. IMPC and IDC tissues shared CNVs of 69 genes, while CNVs of 43 and 5 genes were specifically associated with IMPC and IDC components, respectively (Fig. 3c and Supplementary Data 6). After comparing the CNVs of IMPC and IDC components in primary tumors, we found that primary IMPC components have far more unique genes than IDC, For example, the copy-number loss of AKT3, CCND1, and gains of MYCN, genes were observed in IMPC (Fig. 3e).

Furthermore, in 14 mixed IMPC-IDC tissues, we prepared and sequenced the metastatic foci in the lymph node (each sample included IMPC and IDC components). Seventy-four CNV genes were shared in IMPC and IDC components of lymph node metastases, and 76 and 24 CNV genes were specifically associated with the IMPC and IDC components, respectively (Fig. 3d and Supplementary Data 6). The CNV map showed that 69 genes were shared in primary IMPC and IDC (61.6% of primary IMPC and 93.2% of primary-IDC total CNV genes), while 74 genes were shared in IMPC and IDC in lymph node metastases (49.3% of lymph IMPC and 75.5% of lymph IDC total CNV genes). Thus, IMPC and IDC accumulate more specific CNVs during metastasis than primary tumors, but the overall CNV burden between primary and lymph node metastasis has no significant difference (Student's t test, IMPC, P = 0.63 and IDC, P = 0.038). Unique CNV profiles were also detected in the metastatic lymph nodes; for instance, losses of the genomic loci corresponding to the MAPK1 and gains of ATM genes were observed in IMPC but not in IDC. The overall CNV characteristics are similar between IMPC and IDC, but IMPC has a significantly higher CNV burden in some chromosomes, such as Chr5, 6, 12, 13, 14, and 20 (Fig. 3e). These findings indicate that, while IMPC and IDC share a substantial number of CNVs, IMPC have more unique CNVs and CNV burden than IDC, including the primary tumors and metastases (Supplementary Fig. 10b, c, Student's t test, primary, P = 0.003, lymph node, P = 0.025).

**TCCS reveals the genomic copy-number evolution of the IMPC and IDC components of mixed IMPC-IDC.** The genomic lineage of individual cell clusters of IMPC and IDC in mixed IMPC-IDC tissues (patient P23) was constructed by performing TCCS. The total number of single-cell clusters in P23 was 38, including 1 cluster from normal control, 22 clusters from primary IMPC, 1 cluster from primary IDC, 8 clusters from metastatic IMPC, and 6 clusters from metastatic IDC; the detailed CNV data are shown in Supplementary Data 7. By analyzing the concordances and differences in the CNVs of IMPC and IDC clones in primary and metastatic tumors, we determined the evolutionary relationship between IMPC and IDC during tumor evolution using MEDALT and visualized by TimeScape[34]. Interestingly, primary IMPC and IDC shared some features, i.e., the gain of AKT3, ELF3, and MDM4 genes (Fig. 4a). Moreover, IMPC evolved from IDC in primary mixed IMPC-IDC tissues, as evidenced by the conformity of the genes with CNVs between IMPC and IDC (Fig. 4b, d). The copy-number loss of MUC1 and CDKNB and gain of BRAD1 and ERBB4 play key roles in evolutionary branch nodes. Furthermore, the loss of the ARHGEF10 and PCM1 genes was a driver event of lymph node metastasis in IMPC. During the process of metastasis, IMPC and IDC tend to have their own specific evolutionary processes, leading to more obvious differences in the genome map. We also performed IMPC-specific CNV gene pathway enrichment analyses of both primary and lymph node metastatic lesions of IMPCs (Fig. 4b, d). Pathways, such as Th17 cell differentiation, myeloid cell differentiation, and lymphocyte activation pathways, were highly related to the lymph node metastasis of IMPC (Fig. 4c, e).

**Copy-number evolution of single-cell clusters in primary IMPC.** Using TCCS, we investigated the CNVs of 360 single-cell clusters from 6 primary IMPC cases (P14, P16, P18, P19, P22, and P23; details are shown in Supplementary Data 5) by extracting genomic DNA from single-cell clusters and performing a WGS analysis while preserving the spatial context of the tissue sections. We employed tanglegrams[35] to construct spatial trees (x and y coordinates) with minimal overlapping connections and calculated the genetic distance from the single-cell cluster CNV data while abiding by the Jensen–Shannon distance measure[35]. Moreover, the correlation analysis of the obtained genetic distance and spatial distance indicated no detectable consistency between the spatial distribution and genetic distance in any of the 6 IMPC specimens tested (Fig. 5a–f). This finding suggests that the spatial distance and genetic distance do not match precisely. Thus, a short spatial distance may have a long genetic distance,

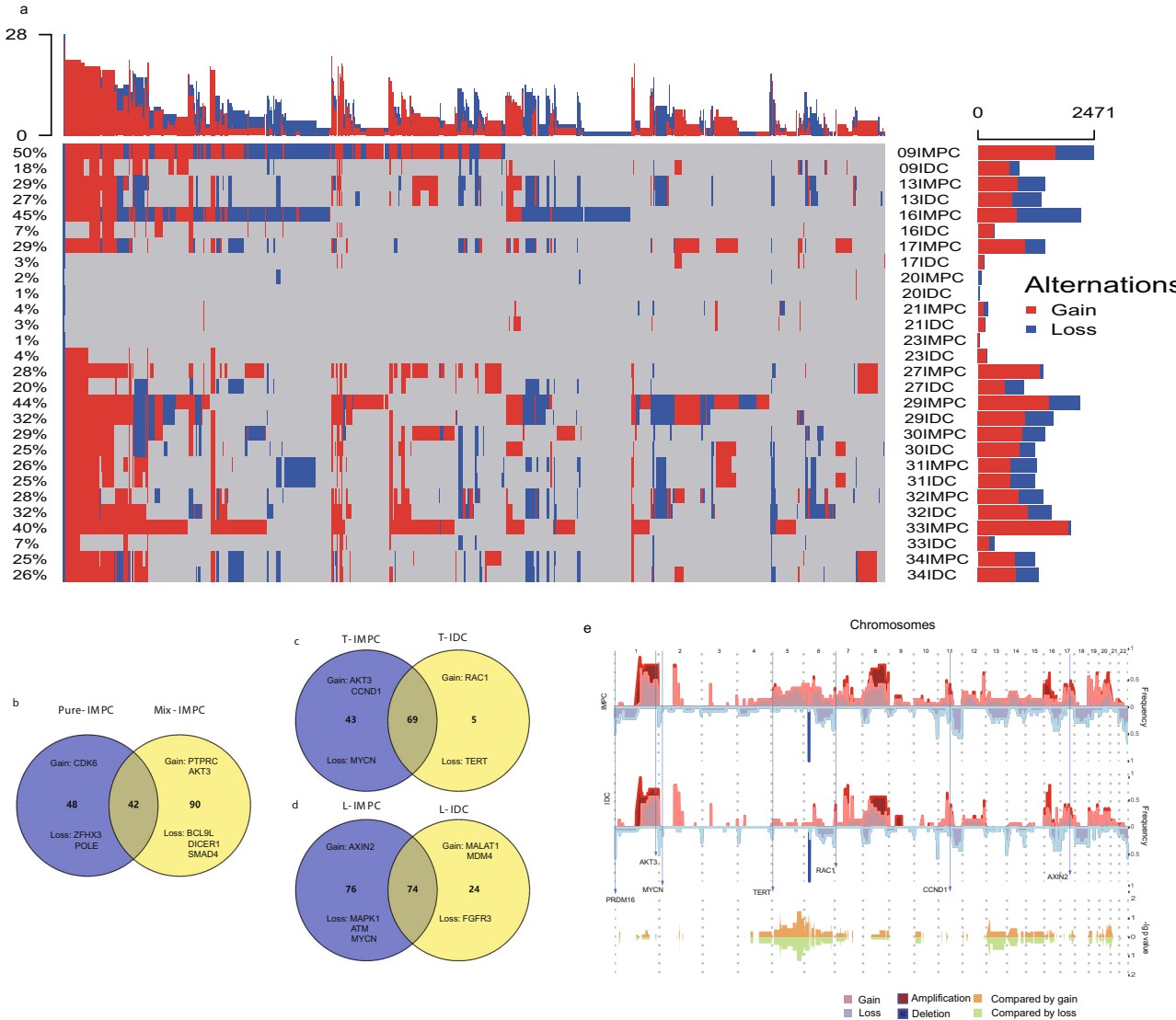

**Fig. 3 Comparison of genomic CNVs in IMPC samples at the level of multiple-cell clusters between pure and mixed IMPC-IDC. a** Genomic CNV landscape in the paired IMPC and IDC components of 14 mixed IMPC-IDC samples. Each row represents a sample; each column represents a genetic locus. The percentages on the left show the genomic instability of each sample across all chromosomal regions, and the bars on the right and top represent the number of different mutations in each sample and the frequency of each genetic locus in the sample, respectively. **b–d** Venn diagrams reflecting common and unique genes with copy-number aberrations in pure IMPC and mixed IMPC-IDC samples (**b**), in IMPC and IDC components of primary tumors (**c**), and in IMPC and IDC components of lymph node metastatic loci (**d**). Common and unique genes with CNVs are labeled in each category. T primary tumor, L lymph node. **e** CNV spectrum of the IMPC and IDC components in mixed IMPC-IDC samples. The frequency of each site was determined by calculating the number of losses or gains in the IMPC and IDC components. The −log10 unadjusted P values are shown for comparison, and significantly different CNV genes are labeled with long arrows. The statistic test is a two-sample unequal variance [heteroscedastic] t test, two-sided.

and a long spatial distance may have a short genetic distance. The developmental lineage of P19 was constructed by hierarchical clustering to study the evolutionary pattern of single-cell clusters in primary IMPC (Fig. 5g); the hierarchical clustering of 5 other IMPC tissues is shown in Supplementary Fig. 6. We constructed a phylogenetic tree using the neighbor-joining method[35] and restored each cell cluster to the corresponding tumor region (Fig. 5h), revealing an inverted pattern of spatial and genetic relationships. In this pattern, some clusters with spatial proximity (e.g., clusters 12 and 15 and clusters 16 and 18) exhibited distant phylogenetic relationships, whereas other clusters displaying long spatial distances (e.g., clusters 11 and 1, clusters 15 and 17, and clusters 19 and 27) exhibited close phylogenetic relationships (Fig. 5h). The tumor cell clusters located outside the tumor mass in IMPC tend to have close genetic distance, which provides a

basis for their unique growth pattern to some extent. These findings enabled us to determine the direction of cell cluster spread in primary tumors.

**Copy-number evolution from primary IMPC to multiple lymphatic metastases**. We analyzed the single-cell clusters of lymph nodes from the same 6 patients with IMPC using WGS to understand the roadmap of single-cell cluster evolution from primary IMPC to metastatic foci. The CNV sequencing data of the primary tumor and single-cell clusters from the lymph nodes of each patient were jointly analyzed. The data from patient P18 are presented as an example in Fig. 6a (the data from the other patients are provided in Supplementary Figs. 7 and 8). Forty-eight single-cell clusters were sequenced, including 24 clusters from the

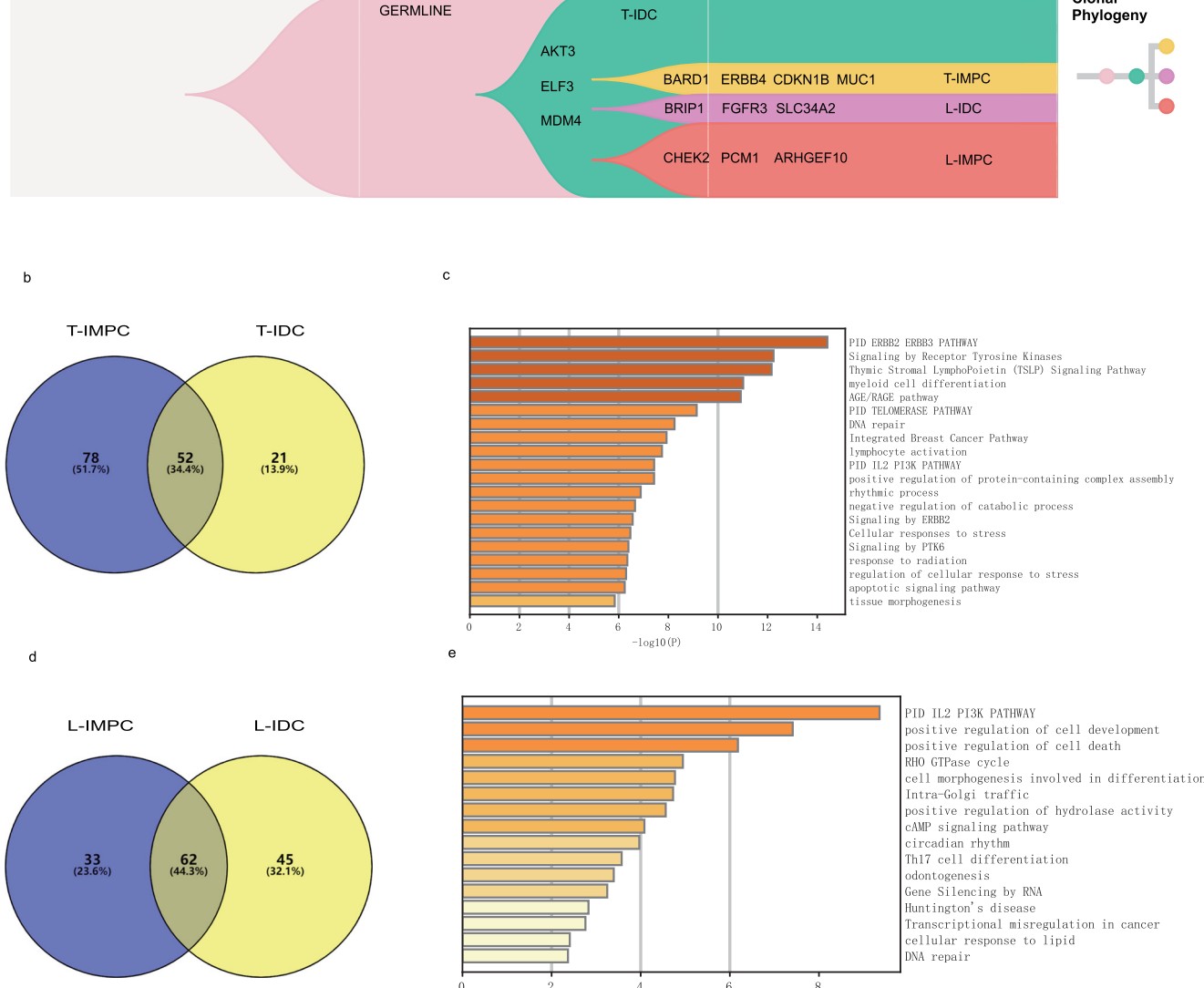

**Fig. 4 Copy-number evolutionary relationship between the IMPC and IDC components in mixed IMPC-IDC. a** Clonal phylogeny in patient P23. Evolutionary relationships between clones were captured by determining the inheritance of direct genomic instability events in different clones. The branching time of the different cell types during the evolutionary process and the respective characteristic driving genes are labeled. T primary tumor, L lymph node. **b** The overlap of CNV genes between the primary IMPC and primary IDC components. Venn diagram showing the distribution of genes with CNVs from the primary tumor. The numbers and percentages of genes in each classification are presented. **c** Pathway analysis of 64 primary IMPC-specific genes. The horizontal axis indicates the significance of the enrichment; the vertical axis indicates the enriched pathways. The well-adopted hypergeometric test and Benjamini–Hochberg *P* value correction algorithm to identify all ontology terms that contain a significantly greater number of genes in common with an input list than expected by chance. **d** Similar to **b**, but an analysis of lymph node metastatic lesions. **e** Similar to **c**, but an analysis of lymph node metastatic lesions.

primary tumor T and the 12, 3, and 9 clusters from lymph nodes L1, L2, and L3, respectively; the three metastatic lymph nodes were randomly named L1, L2, and L3. We constructed a CNV heatmap that depicted the genomic diversity among these forty-eight single-cell clusters (Fig. 6a) and identified the following three clonal aneuploid tumor subpopulations: T, L1, and L2–L3 (Fig. 6a). Furthermore, the CNV dynamics within each sub-population were highly correlated, representing stable clonal expansion. A comparison of the CNVs from the three sub-populations showed that 129 CNV genes were shared by primary and lymph node metastatic IMPC, 43 CNV genes were unique to primary tumors and 30 CNV genes were unique to lymph node metastases. Overall, significant CNV differences existed between the primary and metastatic tumor samples (Supplementary Fig. 9A).

We constructed phylogenetic trees for P18 using MEDALT and visualized by Cellscape[34] to delineate the copy-number evolution of single-cell clusters during metastasis (Fig. 6a). Regarding the CNV evolution, a single-cell cluster in the primary tumor was identified as an ancestor cluster to the metastatic lymph node (L2). Based on these data, independent lymph-node metastases mainly originate from two single-cell clusters and continue to evolve during metastasis. The finding was also verified in the patients P19 and P22 (Supplementary Figs. 7 and 8). The metastatic path of IMPC single-cell clusters is shown in Supplementary Fig. 7C.

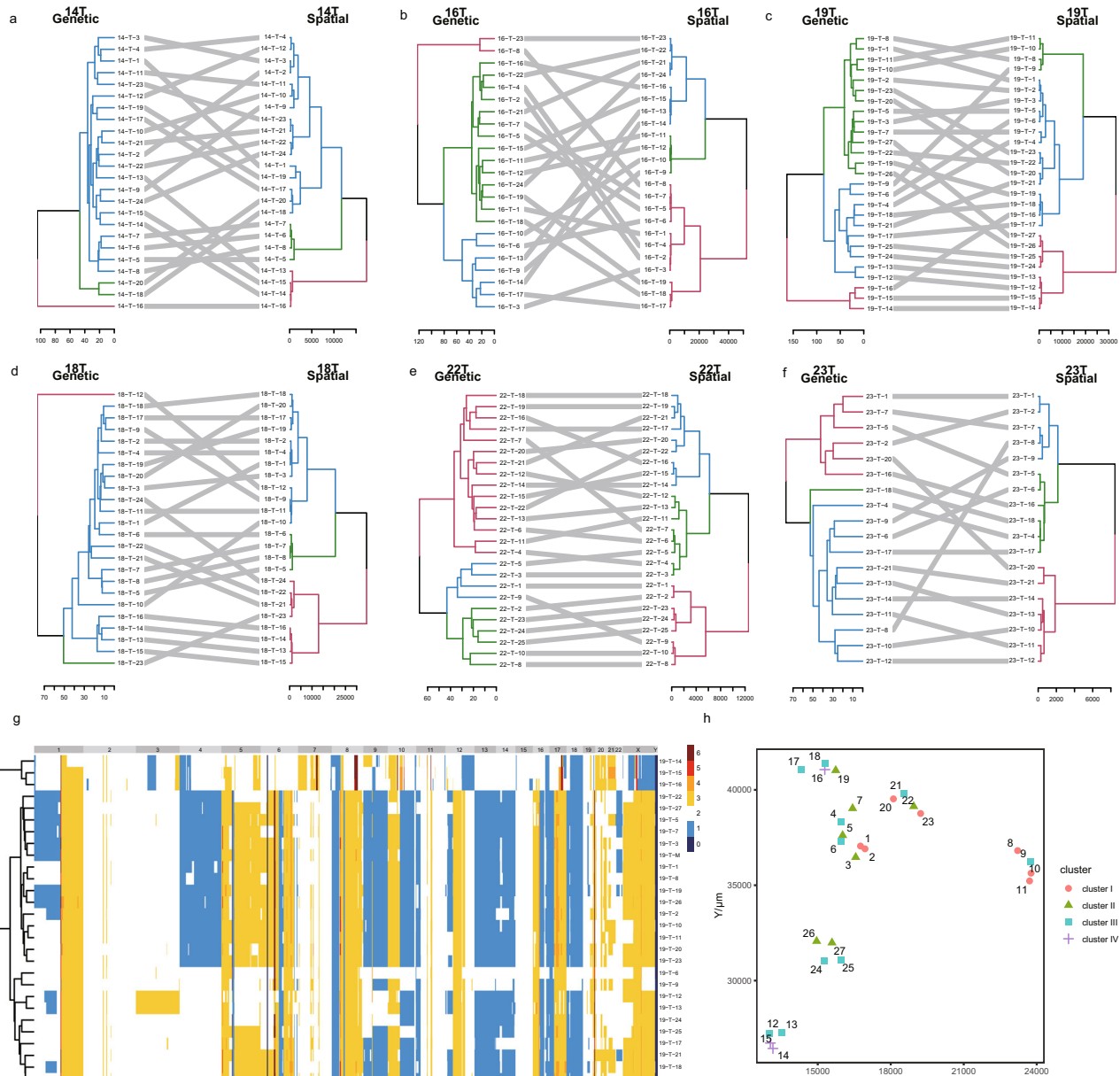

**Fig. 5 Mapping of the spatial and genetic distances of single-cell clusters in primary tumors from six IMPC patients. a–f** Genetic distance and spatial distance mapping tree of primary IMPC (P14, P16, P18, P19, P22, and P23). The spatial distance was obtained by establishing a coordinate system. The genetic distance was calculated by comparing the number of CNVs. The genomic tree is located on the left, and the different colors represent different clonal subpopulations. The spatial tree is located on the right, and the different colors represent the different regions. The mapping of cell cluster coordinates and genotypes was performed by minimizing the overlapping connections. **g** Heatmap of the copy-number profiles of single-cell clusters in P19. Unsupervised clustering based on CNV similarity between different single-cell clusters and the different evolutionary stages during which the cell clusters were isolated is indicated. The values on the right represent the copy number: ≤1 indicates a loss and ≥3 indicates a gain. **h** Tumor section pattern map of P19. A coordinate system for each slice was established to obtain the *x*-axis and *y*-axis information, and the spatial positional relationship of the cell clusters was extracted accordingly. The corresponding evolutionary period of the cell clusters is based on the clustering information.

Two single clusters may belong to the same subclone (monoclonal metastatic seeds) or two different subclones (polyclonal metastatic seeds) in primary tumors, in order to verify the evolutionary paths discussed above, we divided cell clusters into different subclones. We speculate that monoclonal or polyclonal metastatic seeds that may be present in primary IMPC have a higher metastatic potential than other subclones with the same genetic background. We tested this hypothesis by selecting 6 patients with IMPC (P14, P19, P23, P18, P16, and P22) and analyzing the single-cell cluster sequencing data from primary tumors and the paired lymph node metastases to identify the

subclones in the primary tumors that are highly similar to the subclones in the lymph node metastases. The T1, T2, T1, T2, T1, and T1 subclones, which were highly similar to the lymph node subclones, were selected from patients P14, P16, P18, P19, P22, and P23, respectively (Fig. 6b). Two lymph nodes originated clusters of P18 belong to the T1 subclone. By analyzing the genomic features of these subclones with high metastatic potential, we found that these subclones have high-frequency recurrent CNV features, such as the copy-number loss of *IGSF9* and *PRDM16* and gain of *ALDH2* (Fig. 6c). Based on these data, CNVs in the *IGSF9*, *PRDM16*, and *ALDH2* genes may be the key

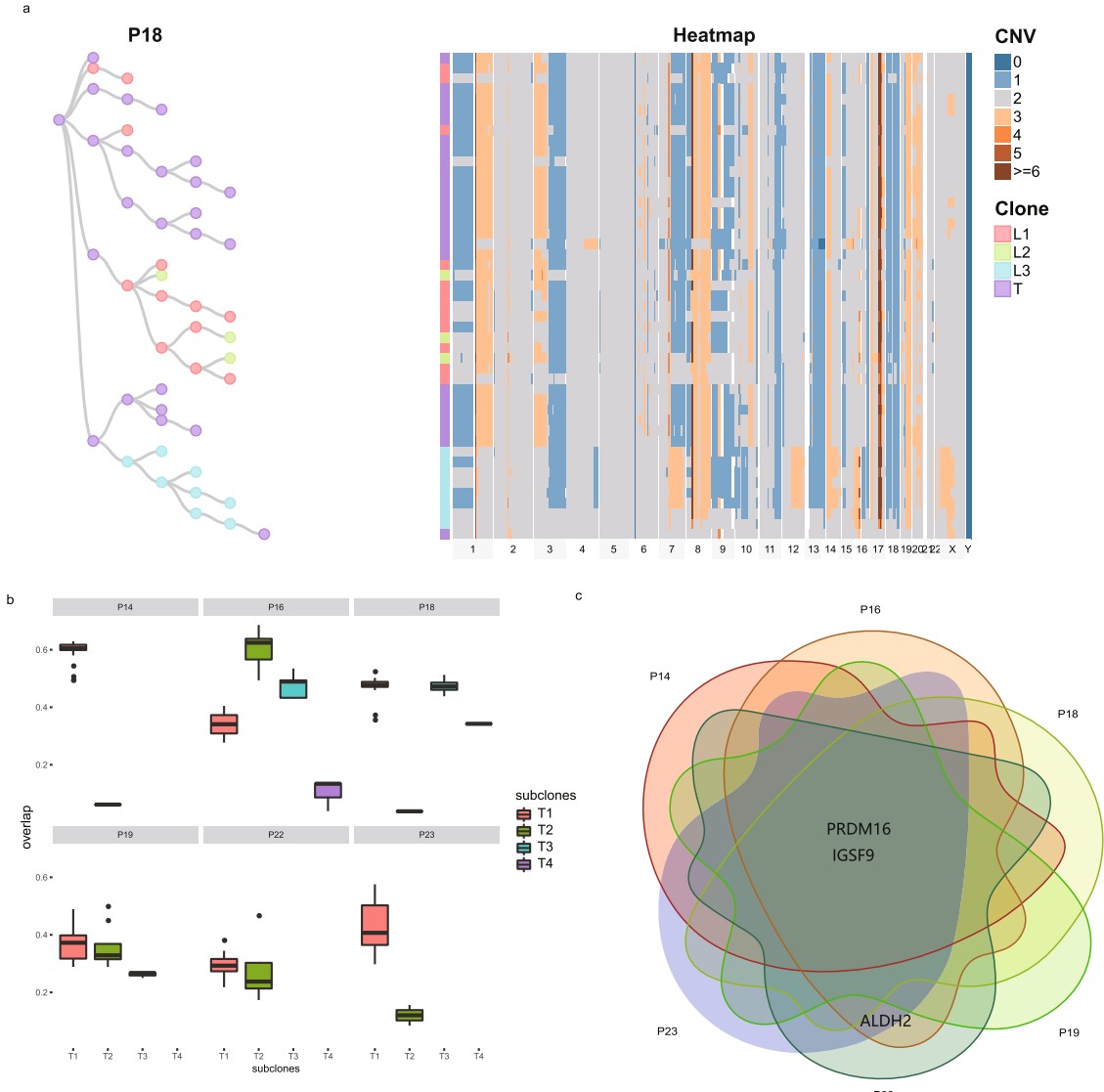

**Fig. 6 Copy-number evolution of single-cell clusters from primary IMPC to metastatic lymph nodes in six samples. a** IMPC single-cell cluster phylogenetic tree of patient P18. Heatmap depicting the genome-wide CNVs (columns) across single-cell clusters (rows). Copy-number gains and losses are encoded by red and blue color gradients, respectively. Tree illustrating the evolutionary relationship based on the minimum spanning tree method. Colored annotations indicate the clonal membership of the cell clusters, showing that multiple lymphatic metastases originated from the same primary subclone. **b** The overlap among the CNV genes in each clone in the primary tumor and lymph node metastases of six patients with IMPC. Boxplots illustrate the degree of similarity between the primary subclone and the metastatic foci at the CNV level in each sample (with the box plot center, box, whiskers, and points corresponding to the median, interquartile range, 1.5× interquartile range, and outliers, respectively). The higher the similarity, the more likely the subclone is to metastasize to lymph nodes; thus, the corresponding single-cell clusters of the subclone have a higher chance of metastasis. The horizontal axis shows the subclone numbers as represented by different colors. The vertical axis shows the similarity rate. **c** Common characteristics of CNVs in subclones selected from **b**. Scaled Venn diagrams reflecting the overlap among subclones from 6 samples with each number representing a CNV shared by two or more subclones; the genes are high-frequency recurrent CNV events among the 6 patients with IMPC, including the loss of *IGSF9* and *PRDM16* and gain of *ALDH2*.

drivers of high lymph node metastasis in patients with IMPC (Fig. 6c). These findings are consistent with the hypothesis that several genes can collectively promote lymph node metastasis when they contain gain or loss CNVs in primary tumors.

We randomly selected 86 FFPE IMPC tissues for the immunohistochemical (IHC) analysis to verify whether these three genes are associated with lymph node metastasis in patients with IMPC. The samples were divided into low- and high-expression groups based on protein expression levels (Fig. 7a–f; see the detailed evaluation criteria in the "Methods" section). The lymph node metastasis status was classified as N0, N1, N2, and

N3 according to the guidelines from the American Joint Committee on Cancer. The correlation analyses showed that IGSF9 and PRDM16 protein levels inversely correlated with lymph node metastasis (IGSF9, $R = -0.32$, $P < 0.01$; PRDM16, $R = -0.336$, $P < 0.01$), while ALDH2 protein expression positively correlated with lymph node metastasis (ALDH2, $R = 0.262$, $P < 0.05$; Supplementary Table 2). Moreover, lower (versus higher) levels of the IGSF9 and PRDM16 proteins were associated with poorer overall survival (OS) (based on the log-rank test, $P < 0.01$ for both) and disease-free survival (DFS) (based on the log-rank test, $P < 0.05$ and $P < 0.01$, respectively), and higher

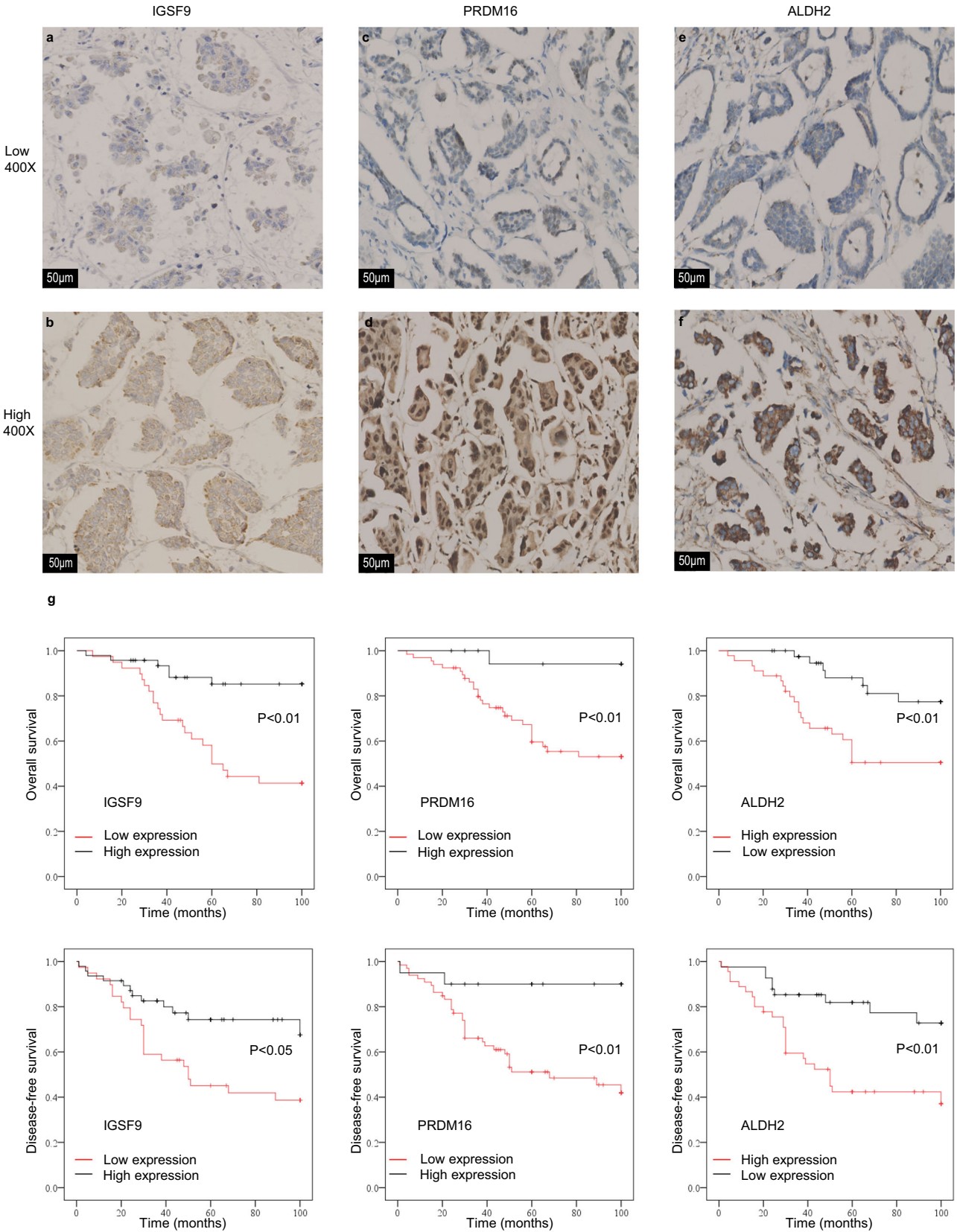

(versus lower) expression of the ALDH2 protein was associated with shorter OS and DFS (based on the log-rank test, $P < 0.01$ for both) (Fig. 7g). Further clinicopathological analyses of these three genes are shown in Supplementary Data 8 and 9. Consistent with the sequencing data, the lower expression levels of the IGSF9 and PRDM16 proteins and the higher expression levels of the ALDH2 protein in IMPC were significantly associated with a higher lymph node metastatic potential.

**Fig. 7 Clinical outcomes associated with the expression of the IGSF9, PRDM16, and ALDH2 proteins in 86 patients with IMPC. a–f** Images of immunohistochemical staining for IGSF9, PRDM16, and ALDH2 in patients with IMPC (*n* = 86). The upper panel shows low expression of IGSF9, PRDM16, and ALDH2 in patients with IMPC; the lower panel shows high expression of the IGSF9, PRDM16, and ALDH2 proteins in patients with IMPC, ×400 magnification. **g** The association of IGSF9, PRDM16, and ALDH2 protein expression with OS and DFS. The upper panel shows OS, and the lower panel shows DFS, with patients with IMPC stratified by high and low protein expression. Log-rank test, two-sided. The *P* values for OS of IGSF9, PRDM16, and ALDH2 were 0.000, 0.004, and 0.004, respectively; the *P* values for DFS of IGSF9, PRDM16, and ALDH2 were 0.010, 0.003, and 0.001, respectively. *P* < 0.05 was considered to indicate statistical significance.

## Discussion

Based on cell differentiation and pathogenesis, invasive breast cancers have been loosely divided into IDC originating from breast ducts and invasive lobular carcinoma originating from breast lobules. IMPC is a variant of IDC. Its tumor cell clusters do not have interstitial connections due to a polarity reversal and the formation of an unstructured gap. Thus, these clusters are able to easily detach and break into single cells or smaller clusters, thereby promoting invasion and metastasis[14]. According to recent studies, clustered tumor cells have a higher metastatic potential than single tumor cells[36–39]. We conducted this study to compare the genomics of primary and metastatic IMPC, to understand why IMPC tumor cells have this unique growth pattern, and to identify the genomic characteristics of the aggressive biological behavior caused by this growth pattern.

Tumor proliferation, invasion, and metastasis are the processes by which cancer cells continuously acquire genetic mutations (SNVs and CNVs). Accumulating evidence has revealed an important role for CNVs at genetic loci in carcinogenesis and cancer progression, including breast cancer[32,40,41]. In 2017, Jamal et al. confirmed that patients with lung cancer presenting more severe CNVs were more difficult to treat and had a worse prognosis[42]. We performed WES and WGS of 17 freshly frozen IMPC samples to obtain a better understanding of the development and metastasis of IMPC and found that the higher lymph node metastatic potential of IMPC was more strongly correlated with CNVs than with SNVs. This finding suggests that studying genomic CNVs is the key to analyzing the mechanism underlying the clustered invasion and metastasis of IMPC cells.

We identified a significant genetic difference in CNVs between IMPC and IDC, even in the same genetic background (same patients). For example, the copy-number losses of *MYCN*, *MAPK1*, and *ATM* were noted in IMPC but not in the IDC component of primary mixed IMPC-IDC. We also found IMPC and IDC components share a large proportion of CNVs. For instance, *PRDM16* functions as a tumor suppressor gene in malignant tumors, and the overexpression of PRDM16 could inhibit EMT by repressing MUC4[43,44], and MUC4 has been reported to be overexpressed in IMPC[45], which is consistent with the losses of PRDM16. In our data, we found losses of PRDM16 is observed in both IMPC and IDC components (Supplementary Data 6). It also exists in the bulk sequencing result (Fig. 2d). In contrast, there are only 1% of patients in TCGA (11/1084) who carried PRDM16 copy-number loss according to online analysis of cBioPortal[46,47], indicating that PRDM16 is the key gene leading to the development of IMPC. And the driver genes of IDC component in IMPC patients may be different from pure IDC patients. Moreover, the differences in CNVs in genes associated with IMPC and IDC in lymph node metastases of mixed IMPC-IDC showed that the copy-number losses of *MAPK1*, *ATM*, and *MYCN* genes were detected in IMPC, but not in IDC. Based on these data, these genes may play a key role in the process leading to a higher lymph node metastatic potential in patients with IMPC. Further analysis revealed that IMPC originated from IDC and that the CNVs in the *MUC1*, *PCM1*, and *ARHGEF10* genes played important roles in the branch node of IDC evolution into IMPC. Among these genes, MUC1 is a marker protein characterized by the polarity reversal of IMPC tumor cell clusters. MUC1 was expressed in the inner luminal surface of IDC glandular ducts and on the mesenchymal side of IMPC tumor cell clusters, leading to the formation of unconnected gaps between IMPC tumor cell clusters and interstitial tissues, which are easily detached. Therefore, the *MUC1*, *PCM1*, and *ARHGEF10* genes play driving roles in the process of IDC evolution into IMPC. The mechanisms and actions of *PCM1* and *ARHGEF10* require further research.

Metastatic subclones in primary tumors are either monoclonal or polyclonal[48,49]. As shown in our study, IMPC may metastatic with monoclonal seeds; i.e., the IMPC cell clusters in different lymph node metastases may originate from one or several single-cell clusters in primary IMPC. These cell clusters in primary IMPC belong to the same subclone, which is the ancestor of IMPC metastasis, metastasizing to the first cell cluster in the lymph node. In the first-line lymph node cell cluster, the ancestor cell proliferates, divides into multiple subclonal cell clusters, and then metastasizes to downstream lymph nodes and even invades soft tissues outside the lymph nodes. During this process, IMPC cell clusters constantly gain CNVs in new genes. Excitingly, the loss of *IGSF9* and *PRDM16* and the gain of *ALDH2* are high-frequency recurrent CNV characteristics of the monoclonal metastatic ancestor in primary IMPC (i.e., these genes are the key driver genes of the high lymph node metastatic potential of IMPC). *IGSF9* is a member of the immunoglobulin superfamily[50] that mainly regulates cell-to-cell adhesion[51]. The downstream effectors and mechanisms of its loss in IMPC require further study. PRDM16 functions as a tumor suppressor in lung cancer[43], and the absence of *PRDM16* in IMPC is consistent with previous findings in lung cancer. Our previous studies confirmed that IMPC cell clusters are rich in stem cells[19]. *ALDH2* is an isoform of aldehyde dehydrogenase (*ALDH*) and has been used as a tumor stem cell marker gene[52,53]. The copy-number gain of *ALDH2* in IMPC indicates that IMPC monoclonal metastatic seeds are rich in stemness, proliferation, and metastatic abilities. The IHC results also showed that the expression of the IGSF9, PRDM16, and ALDH2 proteins was significantly associated with a higher level of lymph node metastasis. Low expression of the IGSF9 and PRDM16 proteins and high expression of the ALDH2 protein indicated a poor prognosis. Their specific functional mechanism and potential use as target genes for the precise treatment of IMPC require further investigation.

In summary, we reported the metastatic genomic analysis of IMPC and examined in detail the repertoire of mutations and copy-number alterations in IMPC. CNVs play an important role in the high lymph node metastatic potential of IMPC and further confirm the evolutionary path of a monoclonal metastatic seed. Furthermore, the copy-number loss of *IGSF9* and *PRDM16* and copy-number gain of *ALDH2* are associated with the expansion of subclones with high metastatic potential and shorter patient survival. These results provide a strong genomic basis and targeted molecules for the precise diagnosis and treatment of IMPC.

## Methods

**Patients and specimens**. The patients and specimens were selected from the full information database of patients with breast cancer in the Department of Breast Cancer Pathology and Research Laboratory, Tianjin Medical University Cancer Hospital, Tianjin, China. The patients included in the studies underwent surgery and systemic treatment at the hospital from 2011 to 2017 and had complete follow-up data, including paraffin-embedded tissue samples, fresh tumor tissue samples, and paired normal breast tissue samples. Furthermore, eighty-six patients with IMPC treated from 2005 to 2010 were selected for IHC validation. The pathological diagnosis was independently determined by two pathologists according to the WHO criteria[3,54] and the IMPC diagnostic criteria proposed in our previous studies[10,13,14]. The morphological characteristics of IMPC tumor cells are provided in Supplementary Fig. 1A, B. All IMPC patients were fully informed and signed informed consent. All procedures performed in studies involving human participants (the samples of patients) were in accordance with the ethical standards of the Ethics Committee of the Tianjin Medical Cancer Institute and Hospital.

**Sequencing of freshly frozen tumor tissues**. We performed WES and WGS using 17 freshly frozen IMPC samples. The details are provided in the sections describing the material preparation and specific methods. The clinicopathological information of the 17 IMPC samples is shown in Supplementary Data 1, and the experimental flow chart is shown in Supplementary Fig. 1C.

**Sequencing of FFPE tumor tissues**. The tumor cell clusters in primary tumors and matched axillary lymph node metastases and the paired normal breast tissues from 29 patients with IMPC (8 with pure IMPC and 21 with mixed IMPC-IDC) were spatially located, coded and microdissected. WGS was performed on the 442 tumor cell clusters obtained from the dissection, and the genomic CNVs were analyzed. The details are presented in the sections describing material preparation and specific methods. The clinicopathological information of the 29 IMPC samples is shown in Supplementary Data 5, and the experimental flow chart is shown in Supplementary Fig. 1D. The specific operation processes are described below.

**Preparation of FFPE tissue sections and IHC staining**. The FFPE tumor tissues of primary IMPC, lymph node metastases, and paired normal breast tissues were continuously sectioned into 4- and 20-μm-thick tissue slices. The 4-μm-thick tissue sections were prepared for hematoxylin–eosin (H&E) staining and used for IHC staining for the estrogen receptor, progesterone receptor, human epidermal growth factor 2, and Ki67 molecules. Additionally, IHC staining for epithelial membrane antigens (ZM0095, ZSGB-BIO) and MUC1 (ab109185, Abcam, 1:1000 dilution) was used to distinguish between IMPC and IDC. The staining for these molecules was assessed according to the American Society of Clinical Oncology/College of American Pathologists guidelines[55,56]. The assessment details are provided in detail in Supplementary Data 1 and 5. IHC was performed on 4-μm-thick tumor sections mounted on slides. Primary antibodies against IGSF9, PRDM16, and ALDH2 (1:250 dilution, HPA037753, Sigma; 1:300 dilution, DF13303, Affinity; 1:250 dilution, ab108306, Abcam, respectively) were used to evaluate the protein expression and were reviewed by at least two pathologists. The IHC staining for PRDM16 was evaluated according to the criteria described by Fei et al.[43], namely according to the percentage of positive tumor cells (1 = 1–25%; 2 = 26–50%; 3 = 51–75%; 4 = 76–100%) and the intensity of staining in the tumor (0 = no staining; 1 = weak; 2 = moderate; 3 = high), the two scores of each tumor sample were multiplied to give a final score of 0–12. If the product of the two scores was ≥6, the PRDM16 was considered highly stained. While IGSF9 and ALDH2 were graded according to the percentage of positive tumor cells (0 = 0%; 1 < 25%; 2 = 25–50%; 3 > 50%) and the intensity of staining in the tumor (0 = no staining; 1 = weak; 2 = moderate; 3 = high), the two scores were multiplied to obtain an overall score. If the product of the two scores was >4, then IGSF9 and ALDH2 were considered highly stained.

**Nuclear fast red (NFR) staining**. The 20-μm-thick tissue sections were mounted onto polyethylene terephthalate (PEN)-coated microscope slides (ABIs) for NFR (Sigma-Aldrich) staining[57]. In order to remove paraffin, the slides were washed with xylene twice for 5 min and then sequentially using 100, 100, 95, 70, 50, and 30% ethanol and nuclease-free water for 5 min to rehydrate. The slides were stained with NFR (Sigma-Aldrich) for 5 min, dehydrated sequentially with 30, 50, 70, 95, 100, and 100% ethanol for 5 min. We compared the H&E- and NFR-stained sections and performed LCM of the tumor cell clusters (including IMPC and IDC) while preserving their spatial positions and morphology in situ.

**Adjustment of the topographic single-cell sequencing method before LCM**. We applied an approach similar to topographic single-cell sequencing[58] to isolate the IMPC and IDC tumor cell clusters from the FFPE tissue sections while preserving their spatial positions and morphology in situ. This approach combines LCM, laser catapulting, whole-genome amplification (WGA), and ultra-low DNA sequencing. After several tests, we were unable to easily identify the IMPC and IDC components in frozen tissue; thus, we modified the procedure for frozen single cells, including the steps that we deemed necessary to successfully analyze a small amount of FFPE tissue including the following: (1) FFPE specimen DNA molecular quality control, (2) removal of the DNA–protein crosslinks, and (3) FFPE DNA damage repair[57].

**FFPE specimen DNA molecular quality control**. We selected the FFPE specimens by applying a multiplex PCR assay[59,60] to define the quality of the DNA extracted from the FFPE samples. This assay uses primer sets that amplify four genomic fragments (100, 200, 300, and 400 bp); the names and sequences of the primers are provided in Supplementary Table 3. Before the multiplex PCR assay, the DNA sample was heated at 90 °C for 60 min to facilitate the removal of the crosslinks of the FFPE tissues, and FFPE DNA repair was performed by adding 1× ThermoPol Buffer, 50x dNTPs/NAD + mixture, and PreCR Repair Mix (NEB) for 30 min at 37 °C[57]. The samples producing 300- and 400-bp fragments were deemed to be good quality and were selected for this study. Ultimately, 29 cases of FFPE IMPC passed the quality control test.

**IMPC and IDC cell cluster morphology and spatial positioning**. H&E staining was performed to generate a global map of the tumor tissue for pathological identification. Using the IHC-stained sections, we identified the IMPC and IDC regions (completed by two pathologists individually) in 20-μm-thick sections that were cut onto PEN-coated microscope slides (Life Technologies). The standard procedure for the NFR (Sigma-Aldrich) staining[57] was described in the above "Nuclear fast red (NFR) staining" part. The tissue sections were scanned under a scanning microscope at ×10 magnification before collection.

**Microdissection of multiple and single-cell clusters from IMPC and IDC**. Whole-tissue sections on slides were scanned and marked as IMPC, IDC, or stroma using H&E and IHC staining. Tumor tissues containing multiple-cell clusters and single-cell clusters were collected using LCM with an ABI system (ABI) while preserving the spatial morphology and location in the sections. Each single-cell cluster was assigned a corresponding number after LCM. Brightfield images were collected before and after tissue capture. The optimal energy for the laser cata-pulting of the tissue ranged from 17–20 delta to reduce DNA fragmentation and increase the collection efficiency. A single-cell cluster refers to one cell cluster. Multiple-cell clusters refer to a collection of several single-cell clusters. The total number of cell clusters was 442, including multiple-cell clusters (82) and single-cell clusters (360 and 164 in primary tumors and 196 in metastatic lymph nodes). Details are provided in Supplementary Data 5.

**Ultra-low nucleic acid WGA**. The tumor cell cluster tissues were laser catapulted into 8 strips of 0.2-mL PCR tubes with 10 μL of lysis solution containing proteinase K from the Sigma-Aldrich Genome Plex WGA4 kit. The tissues were digested for 16 h at 56 °C in the 0.2-mL PCR tube containing lysis buffer and then at 90 °C for 10 min to facilitate the removal of the crosslinks present in the FFPE tissues. After crosslink removal, FFPE DNA repair was performed before WGA by applying a DNA repair enzyme (NEB) for 30 min at 37 °C[57]. The repaired DNA obtained from the previous step was amplified using degenerative-oligonucleotide-primer PCR (DOP-PCR) following the single nucleus sequencing protocol described in previous studies[57,58]. 1× Single-Cell Library Preparation Buffer and Library Sta-bilization Solution were added to samples and mixed well and placed in a ther-mocycler at 95 °C for 2 min. Then the mixture was cooled to 4 °C, mixed with Library Preparation Enzyme, then the thermal reaction was performed at 16 °C for 20 min, 24 °C for 20 min, 37 °C for 20 min, 75 °C for 5 min. At last, 10× Ampli-fication Master and WGA DNA Polymerase were added in the mixture and PCR was conducted as follows: 94 °C for 3 min, 25–27 cycles of 94 °C for 30 s, and 65 °C for 5 min, the PCR cycles were set to 25 for multiple-cell clusters and 27 for single-cell clusters. For quality control, the WGA DNA size distributions were determined through electrophoresis and only samples with fragment sizes >250 bp were selected and purified. The purified WGA DNA was assessed using a Qubit 2.0 Fluorometer (Invitrogen), and samples containing >300 ng of DNA were selected for the library construction and next-generation sequencing.

**Barcoded library construction of amplified DOP-PCR products**. Three hundred nanograms of DNA-amplified DOP-PCR products that passed quality control were fragmented into 250 bp using the Frag enzyme. Qualified genomic DNA was constructed according to the instructions from BGI. Subsequently, pre-hybridization for 95 °C 5 min, 65 °C hold, followed by hybridization for 65 °C 24 h. After elution, 44 μL of products was obtained. A post-PCR reaction mixture including 100 μL of 2× KAPA HiFi HotStart Ready Mix, 6 μL of Ad-153-F (20 μM), 6 μL of Ad-153-R (20 μM), and 44 μL of NF-H2O was prepared and added to the above 44 μL of products. Eleven cycles of PCR amplification were used, and the insert size distributions of the pooled multiplexed libraries were measured using a Bioanalyzer 2100 (Agilent). Subsequently, the samples were modified with splint circulation and ultimately made into a single-strand circular DNA, which resulted in the final library. The rolling circle amplification was performed to produce DNA Nanoballs and then loaded on the BGISEQ-500 sequencing platform. The multi-plexed libraries were sequenced for 110 cycles using paired-end flow cell lanes with a BGISEQ-500 (BGI).

**WES and WGS**. The extracted genomic DNA was fragmented by ultrasound using a Covaris instrument according to the manufacturer's instructions (Covaris Inc., Woburn, MA, USA). The DNA libraries were constructed using end-repair, A-tailing, adapter ligation, and PCR amplification. The WES library was hybridized onto commercial exome capture arrays (BGI Exome V4 Library Kit) for enrichment; the WGS library was not processed. The resulting DNA libraries with an average insert size of 300 bp were subjected to 100-bp pair-end sequencing using a BGISEQ-500 sequencer according to the manufacturer's protocol.

**Somatic mutation analysis**. Raw reads containing adapter sequences and low-quality reads with too many Ns (>10%) or low-quality bases (>50% bases with quality <5) were filtered. The effective reads were mapped onto the hg19 reference human genome using Edico and realigned with the Genome Analysis Toolkit (GATK; v3.4, http://www.broadinstitute.org/gatk). The SNVs were called using Mutect (v1.1.4), and the germline mutations were discarded by filtering based on data from the paired normal sample. ANNOVAR, COSMIC and dbSNP build135 were used to annotate the genes with somatic mutations. The mutation statuses of 978 breast cancers tissues were derived from the Broad Institute TCGA Genome Data Analysis Center (2016): Mutation Analysis MutSig 2CV v3.1. The mutation statuses of 192 breast cancer cases annotated as the luminal B subtype (PAM50[61]) from TCGA database were extracted.

**Copy-number calculations**. For cell cluster sequencing data, clusters meet following criteria is filtered: MAPD > 0.4 or mapping rate < 70% or depth < 0.1 or coverage < 1%. Then the aligned reads of qualified samples were converted from SAM files into BAM files and then sorted using SAM tools (0.1.16). The PCR duplicates were marked and removed using SAM tools. The unique normalized read counts were segmented using the circular binary segmentation method which starts with the whole chromosome and segments it recursively by testing for change-points and stops when none can be found in any of the segments. The algorithm was described in the R Bioconductor "DNA copy" package[62] followed by merging the levels to join adjacent segments with non-significant differences in segmented ratios. The sequencing data were processed according to the "variable binning" pipeline[63,64]. The default parameters were used to merge the levels, which removed erroneous chromosome breakpoints. Control-FREEC software was used to detect the copy-number changes in bulk freshly frozen samples, and GISTIC2.0 was used to identify the regions of the genome with significant gains or losses across all bulk samples. We identified the copy-number changes following two principles. First, we defined a copy number ≤1 as a copy-number loss, while a copy number ≥3 was defined as a copy-number gain, and the copy number in between was defined as normal. Second, for better precision, five CNV states were used according to the results of GISTIC2.0 and the recommended threshold calculated by the GISTIC2.0 algorithm: deletion (log ratio < −1.3), loss (−1.3 ≤ log ratio ≤ −0.1), normal (−0.1 < log ratio < 0.1), gain (0.1 ≤ log ratio ≤ 0.9), and amplification (log ratio > 0.9).

**Gene function enrichment analysis**. Metascape (http://www.metascape.org) was used for the pathway enrichment analysis to investigate the distribution of the genes affected by CNVs within the Kyoto Encyclopedia of Genes and Genomes (KEGG) and Gene Ontology (GO) databases. Enrichment was determined to be informative if the adjusted $P$ value was ≤0.01 (Benjamini–Hochberg method).

**Identification of significant driver genes**. The genomic landscape of IMPC and IDC was depicted using the R package OncoPrint. Then, specific functional differentially expressed genes were selected for a survival analysis using the breast cancer data in TCGA. Survival analysis was conducted using the coxph function in the R package survival version 2.30. Significance was assessed using the likelihood ratio and Wald tests.

**Mapping spatial coordinates and genomic data**. Each sample section area was taken as a unit, Euclidean distance was calculated based on the spatial coordinates of the cell cluster, the lineage was calculated using "ward.D2" in R by the spatial distance. Meanwhile, the phylogenetic tree was based on the genetic distance calculated by CNV inference. Tanglegram version 1.5.2 of DendExtend software package was used to map the genetic tree and spatial tree, and the minimum test set was set to 100 times and gradually extended to reduce the influence of artificial branching.

**Clonal and phylogenetic reconstruction**. The annotated results were filtered using the list of high-frequency genes in cancers in COSMIC to obtain a list of cancer-related mutations in the sample. A pseudotime evolutionary tree of somatic mutated genes was built with MEDALT[65], showing the driver gene in each branch. According to the calculated intercellular evolutionary distance, the R package CellScape was used to construct a single-cell cluster phylogenetic tree with the minimum spanning tree algorithm, and the corresponding drive genes were labeled at each evolutionary branch node. Clonal phylogeny was generated using the R package TimeScape based on the subclone clustering results, and the proportions of different subclones corresponding to cells in each time period were marked. Default parameters were used for all analyses in this part.

**Statistical analysis**. All statistical analyses were performed using R (http://www.r-project.org) and SPSS version 19.0 (SPSS Inc., Chicago, IL, USA). Box plots were generated using the R Base package with the default parameters. Two-sided paired or unpaired Student's $t$ test and the unpaired Wilcoxon rank-sum test was applied as indicated. The clinicopathological characteristics were compared using Mann–Whitney $U$-test. Correlations were analyzed using Spearman rank test. OS and DFS were calculated from the date of surgery to death and locoregional recurrence and/or distant metastasis, respectively. The follow-up time ranged from 4 to 100 months. The curves of OS and DFS were plotted using the Kaplan–Meier method, and the differences between the curves were evaluated using the log-rank test. Univariate and multivariate Cox regression models were applied to analyze the predictors for OS and DFS. $P < 0.05$ was considered to indicate statistical significance.

**Reporting summary**. Further information on research design is available in the Nature Research Reporting Summary linked to this article.

## Data availability

The WES and WGS raw data of frozen tissue and cell clusters have been deposited in European Genome–phenome Archive (EGA) hosted by the EBI and CRG under accession number EGAD00001007588 and in CNGB Nucleotide Sequence Archive (CNSA) under accession number CNP0000395. Those human-related data are accessed via application to Data Access Committee for research purposes. Potential users will need to complete and be approved of a data access request and then the raw data can be used according to the terms of the consent and the data use limitations for the subjects.

## Code availability

R and other custom scripts used to analyze the data are available upon request.

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

## Acknowledgements

We thank all the participants included in this study. This study was supported by the grants from the National Natural Science Foundation of China (Nos. 81672637, 81872164, and 31870860); Guangdong Enterprise Key Laboratory of Human Disease Genomics (No. 2020B1212070028); Joint Fund of the National Natural Science Foundation of China and Natural Science Foundation of Guangdong Province (U1601224); Science, Technology and Innovation Commission of Shenzhen Municipality (No. GJHZ20170314152701465); Shenzhen Key Laboratory of Genomics (No. CXB200903110066A); and Shenzhen Key Laboratory of Single-Cell Omics (No. ZDSYS20190902093613831).

## Author contributions

Q.S., H.W., H.J., K.S., S.L., G.L., and L.F. designed the study. Q.S., S.D., M.L., H.J., and L.F. collected and prepared the samples. Q.S., H.W., K.S., B.C., S.D., H.J., K.W., J.L., J.L., W.L., F.L., S.L., G.L., and L.F. collected the data. Q.S., H.W., K.S., B.C., H.J., H.Z., F.G., S.Z., S.L., G.L., and L.F. analyzed and interpreted the data. Q.S., H.W., K.S., F.G., B.C., L.L., S.L., G.L., and L.F. wrote the manuscript. Q.S., H.W., K.S., B.C., L.L., S.L., G.L., and L.F. revised the manuscript. All authors have reviewed the manuscript and approved the final version.

## Competing interests

The authors declare no competing interests.
