## [Peer Review File · Nature Communications]

REVIEWER COMMENTS

Reviewer #1 (Remarks to the Author):

In this paper, the authors describe the genomic landscape of 17 fresh frozen cases of IMPC, a rare variant of breast cancer characterized by a peculiar growth pattern and a propensity for lymph nodal metastasis.

In addition, they conducted a topographic cell cluster sequencing (TCCS) of individual cell clusters from primary tumors and matched metastatic lymph nodes in FFPE IMPC specimens to determine CNV profiles.

Overall, this represents one of the few genomic studies on this entity, and may contribute to existing literature on IMPC.

However there are major issues regarding this study.

1) The description of the genomic landscape can be much improved. In the description of the mutation landscape, the authors included synonymous (silent) mutations in the main result. These perhaps do not contribute to the pathogenesis and ought to be removed.

The authors describe the mean mutation number per sample - does this include both synonymous and non-synonymous mutations? What was the estimated tumor mutation burden per sample? The majority of the mutations were C > T. Did the authors look at COSMIC mutational signatures in the IMPCs?

2) In general, the authors made several inferences on data comparisons without support from statistical tests.

For example, that PIK3CA mutations in IMPC (11.76%; 2 of 17) compared to breast cancers on TCGA database (31.99%; 311 of 972) is "much lower", is not supported by statistic significance ($p = 0.1116$).

Previous reports (Dieci et al., Breast Cancer Res Treat 2016; Flatley et al., Hum Pathol 2013; Gruel et al., Breast Cancer Res 2014) have demonstrated high frequencies of TP53 and PIK3CA mutations - resembling the mutation profiles of common breast cancers.

Certainly the following conclusion that "Our findings suggest that overall, the mutation spectrum in IMPC significantly differs from that in breast cancer based on TCGA database" cannot be substantiated.

3) In the CNV analysis, "...we found that the CNV distribution in the 17 IMPC samples could be clustered into a deletion group and amplification group with significant differences ($P=0.0045$)" Are there any significant genes found in the specified CNV regions?

4) "Interestingly, the IMPC cases in the amplification group were associated with a substantially higher rate of lymph node metastasis than the cases in the deletion group with invaded lymph nodes percentages of 50.83% and 36.04%, respectively."

It is unclear from Figure 2B which is the "amplification group" and which is the "deletion group". If I interpret the figure correctly, the "amplification group" with 6 cases (P22 to P39) all were node-positive (N1 to N3), as were almost most cases in the "deletion group" with 11 cases (P11 to P45).

The results presented "...invaded lymph nodes percentages of 50.83% and 36.04%" are therefore unclear, and the authors should define how they derived these percentages. No statistics were also presented for this analysis.

5) "The result showed that the highest correlation coefficient between the potential of lymph node metastases and SNVs was 0.46 when using a subclonal population threshold of N=5, whereas the correlation coefficient between lymph node metastases and the CNV levels was 0.54 (0.54 > 0.46). These analyses suggest that the potential of lymph node metastasis is more significantly linked to CNVs than SNVs."

- Do "lymph node metastases" refer to any involvement of lymph nodes, or the number of nodes involved?

6) "IMPC and IDC accumulate more specific CNVs during metastases; the CNV map illustrated that 58 genes are shared by IMPC and IDC in primary tumors (49.2% of primary IMPC and 41.7% of primary IDC among all CNV genes) and that 37 genes are shared by MPC and IDC in lymph node metastases (23.3% of lymph-IMPC and 31.6% of lymph-IDC among all CNV genes)"

- Again no statistical analysis presented to substantiate the conclusion that "IMPC and IDC accumulate more specific CNVs during metastases"

7) From the methods section, WGS was performed on 424 tumor cell clusters. It is not clear how many were "multicell clusters" and how many were "single cell clusters". How many cells were considered "multicell", and what does a single cell "cluster" mean? The exact numbers / tumor sites sequenced per sample should also be summarized and presented.

8) By analysis of subclones in 6 patients from the sequencing of several "single cell clusters", the authors propose that IGSF9, PRDM16 and ALDH2 gene CNVs may be key drivers of high lymph node metastasis in IMPC. Could these CNVs be consistently detected in the earlier analysis using bulk frozen tissue and multicell clusters as well, both in the primary tumor and lymph node metastases?

9) In the discussion the authors claim that "The immunohistochemistry results of the three key metastatic genes, i.e., IGSF9, PRDM16 and ALDH2, in clinical samples of IMPC and IDC have also been verified at the protein level.". Presumably this involved IHC of IMPC and IDC FFPE cases. However, no such results were shown? This is important as the 6 discovery cases is very small in number.

Some minor issues

1) The overall language of the paper ought to be improved.

2) Figures can be better presented e.g. most axes are unlabelled

3) Redundant information in Tables (e.g. Table S1 and S4) can be removed/trimmed. It is also confusing that there are more patients in Table S4 and in Table S1.

Reviewer #2 (Remarks to the Author):

The authors present a genomic study of invasive micropapillary carcinoma (IMPC), a relatively rare breast cancer histological type of breast cancer characterized by an unusual growth pattern with clusters of cells presenting abnormal polarity. The genetic alterations underlying this particular tumor type are still mostly unknown.

Whilst the dataset that the author collected is quite remarkable, especially the sequencing of several tumor cell clusters from matched primary tumors and lymph nodes metastasis, there's a lack of depth in the analyses, which reduces the impact of the findings. Copy number data is the main focus of the study and even though a selection of genes is identified, that may be associated with metastatic potential, further validation is necessary. While functional validation of the candidate role in favoring metastatic potential is probably beyond the scope for this study, the authors should at least consider analyzing publicly available datasets to confirm the potential for the selected genes to drive metastasis, or at a minimum, discuss their findings within the context of previous publications that have explored CNV profiles linked to IMPC (i.e. Natrajan et al. 2014, Marchio' et al. 2008, etc.). The authors conclude that their study 'fundamentally explains the genomic characteristics and evolutionary process of IMPC clustered growth, proliferation, invasion and metastasis'. While the data they generated as part of the study certainly has the potential to explore these essential questions, the presented manuscript does not appear to have conclusively resolved any of the above.

General comments.

- One major criticism is that WES data is analyzed very superficially and immediately discarded as not relevant to tumor invasion and metastasis on a disputable basis. A better description of the mutational patterns of the studied tumors is necessary to conclude whether point mutations do really play a minor role. Also, SNV data would certainly help corroborate the evolutionary paths described in the last result paragraph.
- In several instances the conclusions reported at the end of a result paragraph are not a clear summary of the analysis reported in the same paragraph. As an example, paragraph 5 concludes that 'the ability to develop a considerable amount of genetic variations in IMPC cells is a driving force critical for its high metastatic potential', whereas the entire section was devoted to the comparison between special and genetic distance. This lack of clarity and linearity in the way data and conclusions are presented requires a lot of effort in trying to understand what the message is and if it is effectively supported by the data.
- Similarly, in several instances it is not immediately clear what a specific analysis is supposed to test or prove. Each analysis should be introduced with a clear hypothesis. For example, in the second result paragraph it is not obvious what the Pyclone method is supposed to highlight. The conclusion that a modest difference in correlation coefficients is indicative of a more prominent role for CNV vs. SNV in determining the potential for lymph-node metastasis is not convincing. Having a clear statement of the expectations, a more detailed explanation of the methodology (i.e. which features were correlated?) and an interpretation of the results would help clarify the finding.
- Overall, there's lack of clarity when reporting p-values: in most cases the statistical test used is not

reported, and in several occasions, it is not clear what exactly is being compared (i.e. result paragraph 3: 'the CNV characteristics are significantly distinct between IMPC and IDC (P=0.037)'). P-values are mentioned in the text but not in the corresponding figures.

- Ideally, all of the CNV data indicating a number of affected genes that are listed as candidate drivers of metastasis and IMPC phenotype should be supported by expression data in the presented dataset or, if this is not feasible due to lack of biological material, across independent published datasets.

More detailed comments.

1. Since IMPC is not a well-known tumor type, a better introduction to the concept of tumor cell clusters would help the reader understand the peculiarity of this tumor type.
2. An introductory figure that describes the entire dataset would be very helpful as it is difficult to follow sample availability and sequencing methodology through the different result paragraphs. Similarly, a final schematic representation of the proposed metastatic path would help clarify the main message of the study.
3. Figure 1B. Simply stating the frequency of single nucleotide substitutions is very limited, especially given the recent flourish of tools to compute mutational signatures that consider the three-nucleotide context of each mutation.
4. Figure 1E: although the text references the TCGA BRCA dataset, including a stat analysis with p.value, the figure does not report TCGA data at all, which is confusing. Also, a robust comparison between the authors' dataset and the TCGA BRCA dataset should at least incorporate ER, PR and HER2 status and make sure that mutation frequencies are only compared between matching sets of samples.
5. Figure 2B. the text refers to a p.value which is not reported in the figure. No stats support. The legend is not comprehensive, for example it is not clear what each row represents (is it a genomic window of a specific size?), nor what the blue/red color scheme represents.
6. In the second result paragraph, the authors establish a classification of IMPCs into two distinct groups, based on their CNV profile. However, these groups are not further explored and none of the subsequent analyses refer back to this classification.
7. Figure 3. The entire paragraph 'CNVs in pure IMPC and mixed IMPC-IDC based on multicell-clusters sequencing' is hard to follow and its corresponding Figure 3 is difficult to interpret. According to the text, the samples represented in 3A should be pairs of IDC and IMPC, this is not highlighted in the figure. Also, this entire paragraph is based on intersecting genes affected by gains or losses in a very limited number of samples. It is hard to believe that these findings have statistical relevance and could be reproduced in a larger dataset. It would certainly be more compelling to present a matched analysis of the IMPC-IDC pairs from the same patient, which instead is missing.
8. The use of amplification/deletion vs. gain/loss is quite confusing and should be clarified and kept consistent.

9. Figure 6. The figure must be annotated more clearly, with labels indicating the sample names in each subset.

10. In the discussion, the authors mention immuno-histochemistry data confirming changes in the protein levels of three candidate genes (IGSF9, PRDM16 and ALDH2), however the data is not presented in any figures and it is unclear what exactly has been done.

Francesca Menghi

Reviewer #3 (Remarks to the Author):

This is an interesting study from Shi and colleagues, examining in detail the repertoire of mutations and copy number alterations in a series of invasive micropapillary carcinoma of the breast. However, the authors should elaborate their results more precise and concise, and prove substantial evidence in order to make solid conclusions.

Major points are listed below,

1. The results regarding to the somatic mutations are rather unclear. Please provide a list of somatic mutations identified in primary IMPC and lymph-node metastasis, which should include mutation type, alternate allele, mutation allele fraction, and pathogenicity status.

2. Only non-synonymous mutations should be considered to contribute to pathogenesis, the top 30 mutant genes should be reanalyzed.

3. The frequencies of TP53 and RYR2 mutations in breast cancer in TCGA are not mentioned in the text, thus it is confusing that the authors conclude that only three genes as common mutant genes in IMPC.

4. The data of which version of TCGA the authors used to compare with those of IMPC. The authors should clarify this information in the methods. What histological types of breast cancer do the authors used for the comparisons?

5. Previous studies have shown that IMPC displays a mutation landscape similar to that of luminal B breast cancers. Further comparisons regarding mutations between IMPC and invasive breast cancers with the same immunohistological profile need to be done before the authors make the conclusion for the first paragraph of the result section.

6. For the CNV results in the second paragraph of the results, the authors write that “the most frequent recurrent changes 8p amplification” is incorrect. Previous studies reported a high frequency of 8q amplification in IMPC, which also appears to be the case in this work, as shown in Figure 2A. “8p, 17p and 22q” should be losses but not deletions.

7. The authors claim two novel IMPC-specific genomic amplified regions in 10q14 and 10q22, what is the frequencies of these two regions in IMPC, and are they associated with any genes or clinicopathological features ?

8. Statistic methods need to be written more clearly.
9. Please rectify the color in the Figure 2B. Every one color should represent gain, loss, amplification and deletion, respectively.
10. It is interesting that the authors elaborate the CNVs of IMPC, however, to conclude that CNVs are more significantly associated with lymph node metastasis rather than SNVs requires more comprehensive analyses.
11. The authors need to distinct gain and loss from amplification and deletion, respectively, throughout the manuscript, including the figures and tables.
12. For results in Supplementary Table 2, "common" means these genes share in all cases? Otherwise, frequencies need to be noted for each gene, as well as for those exclusively altered genes, and then define the recurrent exclusively altered genes in both pure-IMPC and IDC.
13. For the results in Supplementary Table 2. Base on the frequencies of gains and losses, statistical analyses should be done to define the significant alterations in each component, rather than simply find the exclusive altered genes.
14. For the third section of the results, I suppose the authors did the comparisons between all 8 pure IMPC and 21 mixed IMPC, as shown in the first sheet of Supplementary Table 2. If so, the gain of PRDM16, PRCC and ELF3 occurs in the IDC component from the eight mixed IMPC cases in the second sheet of Supplementary Table 2 should also appear in the first sheet in the exclusively in mixed-IMPC column. The number of cases that subjected for sequencing should be noted.
15. The authors identified deletions of TNFRSF14, MYC and PCM1 as truncal CNVs in primary IMPC and IDC. As far as I know, MYC is a proto-oncogene, while the authors find deletion of MYC as a truncal event in the primary IMPC and IDC really baffles me.
16. MUC1, MAPK1 and ARHGEF10 can not be found in the first sheet of Supplementary Table 2, please clarify the type of alterations of these three genes.
17. The functions of PRDM16 and IGSF9 are rarely investigated, especially in breast cancer, it would be interesting if the authors could engage some functional studies to prove the driven effects of these genes in IMPC metastasis.
18. It is difficult for me to understand why IGSF9 and ALDH2 gene CNVs were not shown in the results of Figure 3 and Figure 4, due to the sequencing depths? As these gene CNVs are common features related to high metastatic potential as the authors claim.
19. The authors mention in the Discussion section that "The immunohistochemistry results of the three key metastatic genes, i.e., IGSF9, PRDM16 and ALDH2, in clinical samples of IMPC and IDC have also been verified at the protein level.", where are these results? If there are, correlations between each gene expression and clinicopathological features and prognosis should be calculated.
20. Supplementary Table 1 and 4 should be merged, whist, there are 30 but not 29 cases in

supplementary Table 4.

21. The authors applied single cell-cluster sequencing data in both primary tumors and metastatic lymph nodes of six cases, and identified the deletion of IGSF9 and PRDM16 and amplification of ALDH2 as drivers for lymph node metastasis in IMPC. Among the six cases, one has both IMPC and IDC component metastases in lymph node that was sequenced. To also show that IGSF9 and ALDH2 are exclusively altered in the IMPC component will solid authors' conclusion.

22. The results need to be reorganized and discussed accordingly and concisely in order to make it easier to follow.

REVIEWER COMMENTS

We thank the editor and the three referees for their insightful comments and constructive suggestions. We have made extensive modifications and corrections accordingly on the manuscript and provided additional supplementary data. The following is a summary of the major revisions:

1. Since IMPC is a rare breast tumor subtype, we have added a few statements in the Introduction section in order to help the readers to understand the peculiarity of this tumor type.

2. In the first paragraph of the Results section, we further analyzed the somatic mutations in 17 pairs IMPC frozen samples, deleted the synonymous mutations (**Figure 1 and Figure S2**), provided a list of all mutated genes (**Supplementary Table 2**), and included TCGA breast cancer gene mutation analysis.

3. In the second paragraph of the Results section, we added more detailed analysis of CNV with the software (Control-FREEC), and provided the updated results (**Figure 2**) to further demonstrate that CNV plays a more important role than SNV does in lymph node metastatic potential.

4. In the third paragraph of the Results section, we adjusted the order of content in order to help the readers to follow.

5. we conducted the immunochemistry for the protein expression of IGSF9, PRDM16 and ALDH2, and reported the results in the last paragraph of the Result section (**Figure 7 and Supplementary Table 8-10**).

6. We have also unified the terminology and used gain/loss to describe the degree of CNVs.

Our point-by-point responses are given below. The comments made by the three reviewers are underlined.

Reviewer #1

In this paper, the authors describe the genomic landscape of 17 fresh frozen cases of IMPC, a rare variant of breast cancer characterized by a peculiar growth pattern and a propensity for lymph nodal metastasis.

In addition, they conducted a topographic cell cluster sequencing (TCCS) of individual cell clusters from primary tumors and matched metastatic lymph nodes in FFPE IMPC specimens to determine CNV profiles.

Overall, this represents one of the few genomic studies on this entity, and may contribute to existing literature on IMPC.

We truly appreciate the positive comments from the reviewer.

However, there are major issues regarding this study.

1) The description of the genomic landscape can be much improved. In the description of the mutation landscape, the authors included synonymous (silent) mutations in the main result. These perhaps do not contribute to the pathogenesis and ought to be removed.

We concur. We have removed the synonymous mutations from the revised manuscript (**Figure 1** and **Figure S2**).

The authors describe the mean mutation number per sample - does this include both synonymous and non-synonymous mutations? What was the estimated tumor mutation burden per sample?

The description of the mean mutation number per sample included both synonymous and non-synonymous mutations. According to the insightful comment, we have removed the synonymous mutations in the analysis. In the revised text, the mean mutation number per sample was 58 (**Table R1-1**). The estimated tumor mutation burden (TMB) per sample is shown in **Figure S2A**. We calculated the TMB for each sample and compared it with the TCGA breast cancer data. No statistical significance was found according to the two-sample unequal variance (heteroscedastic) *t*-test ($P = 0.178$).

IMPC	nonsys	TMB(nonsys/CDS)
Media	58.00	1.71
mean	79	2.32
TCGA 978	nonsys	TMB(nonsys/CDS)
Media	28.00	0.83
mean	59	1.75
TCGA LumB	nonsys	TMB(nonsys/CDS)
Media	34.00	1.00
mean	54.86	1.62
Compare type	t.test p value	nonsys mutation num
IMPC-VS-TCGA 978		0.177705801
IMPC-VS-TCGA LumB		0.108369917
TCGA 978-VS-TCGA LumB		0.649119924

Table R1-1: The comparison of the mean tumor mutation burden (TMB) of IMPC, TCGA breast cancer and TCGA luminal B subtype breast cancer. Nonsys, non-synonymous. TMB = number of non-synonymous mutations / length of CDS. P-values were calculated by the two-sample unequal variance (heteroscedastic) *t*-test. P < 0.05 was considered to indicative of statistical significance.

The majority of the mutations were C > T. Did the authors look at COSMIC mutational signatures in the IMPCs?

This is a great point, which we highly appreciate. Since IMPC is a special subtype of breast cancer, there is no separate data classification for IMPC in COSMIC presently. In addition, in the literature, C > T mutations are considered a common event in breast cancer and other epithelial tumors (Dieci *et al.*, Breast Cancer Res Treat 2016), which is consistent with the mutational signature distribution in TCGA (**Figure R1-2**). Thus, we do not think this feature can be used as a unique feature of IMPC. Furthermore, only 17 samples of WES data were available, we used the original nonnegative matrix factorization (NMF) signature decomposition method (Nik-Zainal S. *et al.*, Cell (2012)) to analyze the mutational signatures and found that the reconstruction error was high and that only the signatures 1 and 2 were stable (**Figure R1-3**). Thus, the number of somatic mutations in the 17 IMPC samples seems insufficient for mutational signatures analysis.

Figure R1-2: Mutational signatures distribution in TCGA breast cancer (978 samples). Updated as **Figure S2B**.

Figure R1-3: IMPC somatic mutation signatures analyzed by NMF algorithm. Each x-axis represents a signature feature. A stability index above 0.9 is considered an accurate signature.

2) In general, the authors made several inferences on data comparisons without support from statistical tests.

For example, that PIK3CA mutations in IMPC (11.76%; 2 of 17) compared to breast cancers on TCGA database (31.99%; 311 of 972) is "much lower", is not supported by statistic significance ($p = 0.1116$).

Previous reports (Dieci *et al.*, Breast Cancer Res Treat 2016; Flatley *et al.*, Hum Pathol 2013; Gruel *et al.*, Breast Cancer Res 2014) have demonstrated high frequencies of TP53 and PIK3CA mutations - resembling the mutation profiles of common breast cancers.

Certainly the following conclusion that "Our findings suggest that overall, the mutation spectrum in IMPC significantly differs from that in breast cancer based on TCGA database" cannot be substantiated.

We thank the reviewer for pointing out this problem. Since we had only 17 frozen samples for bulk sequencing, which is not sufficient for statistic tests. We therefore deleted the word “significantly” in the statement. Additional changes were made to soften the assertive tune or show the used statistical test.

3) In the CNV analysis, "we found that the CNV distribution in the 17 IMPC samples could be clustered into a deletion group and amplification group with significant differences (P=0.0045)"

Are there any significant genes found in the specified CNV regions?

Thanks to the reviewer’s invaluable suggestion, we performed more detailed analysis of CNV. We found that the previously selected software (FACET) is more inclined to obtain large fragments of CNV regions and appeared unable to analyze gene-level changes. To overcome this shortcoming, we employed the Control-FREEC analysis software (**Figure R1-4**). The results were slightly different, and the main conclusions remained the same. The relevant modifications have been made to the text (**Figure 2**).

Based on the new high-resolution CNV data, several special copy number alteration regions were identified, including gain peaks containing some known breast cancer driver genes such as *ERBB2* and *NOTCH2* and loss peaks containing known tumor suppressor genes such as *PRDM16* and *STK11* (**Table R1-5**, more details are provided in **Supplementary Table 3**).

Furthermore, in the revised manuscript, we refer to the two groups as the CNV high group and CNV low group instead of the amplification group and deletion group, respectively (**Figure 2**).

Figure R1-4: Comparison of the software before (FACET) and now (Control-FREEC). Control-FREEC can obtain more small fragments of CNV to analyze significant genes. IchorCNA was utilized to identify technical noise in the copy number profiles.

Type	chr_arm	q value	gene in CGC_v80
Gain	11q13.3	0.00022415	CCND1
Gain	17q12	9.70E-05	ERBB2,CDK12
Gain	17q23.2	0.0015224	BRIP1
Gain	8q23.1	0.027878	EXT1,EIF3E,RAD21,RSPO2
Gain	20q13.2	0.068745	NA
Gain	10q22.3	0.1313	NUTM2B
Gain	1q21.2	0.15119	BCL9,NOTCH2,PDE4DIP
Gain	8p11.23	0.15561	NA
Gain	6p25.2	0.18456	NA
Gain	11p12	0.20079	NA
Loss	5q11.2	0.0054922	IL6ST,MAP3K1
Loss	8p21.3	0.0054922	PCM1
Loss	19p13.3	0.0054922	STK11,TCF3,FSTL3
Loss	1p36.32	0.0054922	TNFRSF14,PRDM16
Loss	16q24.1	0.040674	NA
Loss	11q25	0.050507	NA
Loss	6q21	0.069935	NA
Loss	13q34	0.079853	NA
Loss	11p15.5	0.096699	HRAS
Loss	18q12.2	0.098789	NA
Loss	2q37.3	0.1451	ACKR3
Loss	10q26.3	0.1451	NA
Loss	16p13.3	0.15241	TSC2,AXIN1,TRAF7
Loss	8q24.3	0.18223	NA
Loss	4q34.3	0.20993	NA
Loss	9q34.11	0.20993	FNBP1
Loss	17q12	0.20993	NA
Loss	4p16.3	0.21608	FGFR3
Loss	14q32.12	0.15525	TCL1A,TRIP11,GOLGA5,DICER1
Loss	20q13.33	0.22781	PTK6,SS18L1
Loss	14q32.33	0.040674	NA

Table R1-5: Significant genes in the specified CNV regions. The cutoff value of q is 0.25 for selecting significant regions.

4) "Interestingly, the IMPC cases in the amplification group were associated with a substantially higher rate of lymph node metastasis than the cases in the deletion group with invaded lymph nodes percentages of 50.83% and 36.04%, respectively."

It is unclear from Figure 2B which is the "amplification group" and which is the "deletion group". If I interpret the figure correctly, the "amplification group" with 6 cases (P22 to P39) all were node-positive (N1 to N3), as were almost most cases in the "deletion group" with 11 cases (P11 to P45).

The results presented "...invaded lymph nodes percentages of 50.83% and 36.04%" are therefore unclear, and the authors should define how they derived these percentages. No statistics were also presented for this analysis.

We concur. In the revised manuscript, we refer to the two groups as the CNV high group and CNV low group instead of the amplification group and deletion group, respectively.

The CNV low group included 9 samples (P11, P17, P36, P43, P42, P19, P38, P44 and P39), and the CNV high group included 8 samples (P45, P8, P22, P41, P37, P40, P35 and P14) (**Figure 2A**).

The mean lymph-node metastasis rate in the CNV-high group and CNV-low group were 57.63% and 26.72%, respectively. The mean lymph node metastasis rates of the new groups were calculated as shown below. In the revised manuscript, we applied the AJCC lymph node staging-category (N0-N3) to describe the lymph node (LN) status. According to the Mann-Whitney U-test, the CNV high and low groups exhibited a significant difference (P=0.045); the details are shown in **Table R1-6**.

Groups	Patient ID	Metastatic LN number	Total LN number	Metastatic rate	Mean LN metastatic rate	LN stage	P-value (Mann-Whitney U-test)
CNV high group	P45	26	30	86.67%	57.63%	N3	0.045
	P8	3	26	11.54%		N1	
	P22	16	21	76.19%		N3	
	P41	12	17	70.59%		N3	
	P37	2	10	20.00%		N1	
	P40	16	22	72.72%		N3	
	P35	15	18	83.33%		N3	
	P14	6	15	40.00%		N2	
CNV low group	P11	3	11	27.27%	26.72%	N1	
	P17	1	23	4.35%		N1	
	P36	3	19	15.79%		N1	
	P43	2	21	9.52%		N1	
	P42	0	15	0.00%		N0	
	P19	16	21	76.19%		N3	
	P38	2	24	8.33%		N1	
	P44	2	15	13.33%		N1	
	P39	18	21	85.71%		N3	

Mean LN metastatic rate:

$$\text{High group} = (26/30 + 3/26 + 16/21 + 12/17 + 2/10 + 16/22 + 15/18 + 6/15) / 8 = 57.63\%$$

$$\text{Low group} = (3/11 + 3/19 + 1/23 + 2/21 + 0/15 + 16/21 + 2/24 + 2/15 + 18/21) / 9 = 26.72\%$$

Table R1-6: Difference between the CNV groups and lymph node metastasis stages. LN, lymph node. The CNV high group has 8 samples, and N3 accounts for 5/8 samples, the CNV low group has 9 samples, and N3 accounts for 2/9.

5) "The result showed that the highest correlation coefficient between the potential of lymph node metastases and SNVs was 0.46 when using a subclonal population

threshold of N=5, whereas the correlation coefficient between lymph node metastases and the CNV levels was 0.54 (0.54 > 0.46). These analyses suggest that the potential of lymph node metastasis is more significantly linked to CNVs than SNVs."

- Do "lymph node metastases" refer to any involvement of lymph nodes, or the number of nodes involved

"Lymph node metastases" refers to the numbers of lymph nodes metastases. In our data, the potential of metastatic potential of IMPC was evaluated by the AJCC lymph node staging-category (N0-N3). The higher the lymph node staging-category is, the more lymph nodes are involved, and the higher the metastatic potential. We have made clarification in the revision.

6) "IMPC and IDC accumulate more specific CNVs during metastases; the CNV map illustrated that 58 genes are shared by IMPC and IDC in primary tumors (49.2% of primary IMPC and 41.7% of primary IDC among all CNV genes) and that 37 genes are shared by MPC and IDC in lymph node metastases (23.3% of lymph-IMPC and 31.6% of lymph-IDC among all CNV genes)"

- Again no statistical analysis presented to substantiate the conclusion that "IMPC and IDC accumulate more specific CNVs during metastases"

We thank the reviewer for the suggestion. The P-values of the conclusion that "IMPC and IDC accumulate more specific CNVs during metastases" are 0.0012 and 0.014, respectively (two-sample unequal variance (heteroscedastic) *t* test).

7) From the methods section, WGS was performed on 424 tumor cell clusters. It is not clear how many were "multicell clusters" and how many were "single cell clusters". How many cells were considered "multicell", and what does a single cell "cluster" mean? The exact numbers / tumor sites sequenced per sample should also be summarized and presented.

We thank the reviewer for the suggestion. The most significant signature of IMPC tissues is the clustered growth pattern. In the present research, a single cell-cluster refers to one cell-cluster (**Figure R1-7** below). Multiple cell-clusters refer to a collection of several single cell-clusters. "Single cell-cluster" and "multiple cell-clusters" were laser-capture microdissection (LCM) isolated and subjected to downstream WGS analysis. We have clarified this point in the Material and Methods section.

The numbers of multiple cell-clusters and single cell-cluster included in study are provided in the following forms (**Figure R1-8**). There is a total of 422 cell clusters, including 63 multiple cell-clusters, 359 single cell-clusters. Furthermore, the numbers of cell clusters in each sample, the tumor sites and the location where each sample appears in the figures are provided in **Table R1-8** and **Table R1-9** and also provided in the **Supplementary Table 5**.

Figure R1-7: Difference between multiple cell-clusters and single cell-cluster. The left is a multiple cell-clusters, and the right is single cell-cluster.

Patient ID	The proportion of IMPC	Numbers of LNM	Primary tumor (IMPC)	Primary tumor (IDC)	LNM (IMPC)	LNM (IDC)	Total (63)	Where it appears
1	100	31/31	1	NA	NA	NA	1	Figure 3B-a
2	90	5/25	NA	NA	NA	1	1	
3	80	4/24	NA	1	1	NA	2	
7	100	4/18	NA	NA	1	NA	1	
8	100	3/26	1	NA	NA	NA	1	
9	50	15/22	NA	1	NA	1	2	
10	100	17/28	NA	NA	1	NA	1	
11	100	3/11	NA	NA	2	NA	2	
12	95	6/22	NA	NA	1	1	2	
13	55	2/18	1	1	1	NA	3	
15	85	16/21	1	NA	1	NA	2	
17	70	1/23	NA	1	NA	NA	1	
20	70	11/25	1	1	NAA	NA	2	
21	40	18/18	1	1	N	NA	2	
24	100	2/11	1	NA	NA	NA	1	
27	40	4/30	2	2	N	1	5	Figure 3A, Figure 3B-b, c, Figure 3C
28	30	16/30	1	2	2	2	7	
29	30	17/23	1	1	1	1	4	
30	60	3/18	1	1	1	1	4	
31	40	1/22	1	2	1	1	5	
32	40	6/16	1	1	N	2	4	
33	70	2/12	1	1	1	1	4	
34	70	21/27	1	1	2	2	6	

Table R1-8: Information of multiple cell-clusters sequencing, including 23 samples and 63 multiple cell-clusters, and the numbers of cell clusters in each sample, the tumor sites and the location where each sample appears in the manuscript figures. LNM, lymph node metastases.

Patient ID	Proportion of IMPC	Numbers of LNM	Numbers of single cell-cluster in the primary tumor (IMPC)	Numbers of single cell-cluster in the primary tumor (IDC)	Numbers of single cell-cluster in LNM (IMPC)	Numbers of single cell-cluster in LNM (IDC)	Total (359)	Where it appears
14	80	6/15	25	NA	2	NA	27	Figure 5A, 5I, 6B, 6C
16	80	2/18	18	NA	19	NA	37	Figure 5B, 5I, 6B, 6C
18	90	3/17	17	NA	33	NA	50	Figure 5D, 5I, 6A, 6B, 6C
19	100	16/21	18	NA	41	NA	59	Figure 5C, 5G, 5H, 5I, 6B, 6C
22	100	16/21	36	NA	111	NA	147	Figure 5E, 5I, 6B, 6C
23	30	5/13	18	6	6	9	39	Figure 4A-E, 5F, 5I, 6B, 6C

Table R1-9: Information of single cell-cluster sequencing, including 6 samples and 359 single cell-clusters, and the numbers of cell clusters in each sample, the tumor sites and the location where each sample appears in the manuscript figures. LNM, lymph node metastases.

8) By analysis of subclones in 6 patients from the sequencing of several "single cell clusters", the authors propose that IGSF9, PRDM16 and ALDH2 gene CNVs may be key drivers of high lymph node metastasis in IMPC. Could these CNVs be consistently detected in the earlier analysis using bulk frozen tissue and multicell clusters as well,

both in the primary tumor and lymph node metastases?

Thank you for the suggestion. We used GISTIC2.0 to detect CNV focal regions in bulk frozen tissue, and found that *PRDM16* in the 1p36.32 is a loss focal region.

The loss of *PRDM16* was significantly different in the CNV high and low groups (**Table R1-10**) (student's *t*-test, $P = 0.011$), indicating that the CNV high group is more likely to enrich this gene loss, which is consistent with the result that *PRDM16* may be a key driver gene of lymph node metastasis in IMPC. In addition, the loss of *IGSF9* (1p23.3) and *PRDM16* (1p36.32) and the gain of *ALDH2* (12q24.33) were detected in multiple cell clusters both in primary tumors and metastases; the details are provided in the updated **Supplementary Table 6**.

Groups	Patient ID	Chr1:35002:30495000 (copy number log ratio)	P-value (student's t - test)
CNV high group	P45	-0.41366	0.011
	P8	-0.14039	
	P22	-0.77048	
	P41	-0.33733	
	P37	0.0076392	
	P40	-0.47014	
	P35	0.096816	
	P14	-0.36397	
CNV low group	P11	-0.078411	
	P36	0.0558	
	P17	0.097397	
	P43	0.12314	
	P42	0.12977	
	P19	0.049114	
	P38	-0.0041546	
	P44	-0.049154	
	P39	-0.29627	

Table R1-10: The CNV of *PRDM16* was significantly different between the CNV high group and low group (student's *t*-test, $P=0.011$).

9) In the discussion the authors claim that "The immunohistochemistry results of the three key metastatic genes, i.e., IGSF9, PRDM16 and ALDH2, in clinical samples of IMPC and IDC have also been verified at the protein level." Presumably this involved IHC of IMPC and IDC FFPE cases. However, no such results were shown? This is important as the 6 discovery cases is very small in number.

Thank you for the comment. We have added the immunohistochemistry results of the three key metastasis genes (*IGSF9*, *PRDM16* and *ALDH2*) in the last paragraph of the

results (**Figure 7**). We validated the three genes in tumor samples of 86 IMPC patients. Furthermore, the correlations between the gene expression level and clinicopathology features are provided in **Supplementary Table 9**. The overall survival and disease-free survival of IMPC patients in the high gene expression group and low gene expression group are provided in **Figure 7G**, and the prognostic factors for overall survival and disease-free survival were added to **Supplementary Table 10**.

Some minor issues

1) The overall language of the paper ought to be improved.

We have invited an editing company, American Journal Experts (AJE) (ID: 9DJ4S4PB), and an English speaker to help polish our article extensively.

2) Figures can be better presented e.g. most axes are unlabelled.

We have checked the entire article and added the labels for the axes, and the details are shown in the figures and legends.

3) Redundant information in Tables (e.g. Table S1 and S4) can be removed/trimmed. It is also confusing that there are more patients in Table S4 and in Table S1.

In the revised manuscript, we merged **Supplementary Tables 1 and 4**, trimmed some redundant information, such as “Gender” and “Age” and created a new updated **Supplementary Table 5**. Furthermore, we thank the reviewer for pointing out this mistake. As P5 failed to pass the DNA sample quality control, 29 IMPC samples were included in present study. However, in the original **Supplementary Table 4**, we enrolled P5 in the IMPC samples and had 30 IMPC patients. We apologize for our mistake. In the revised **Supplementary Table 5**, we have corrected this error.

Reviewer #2

The authors present a genomic study of invasive micropapillary carcinoma (IMPC), a relatively rare breast cancer histological type of breast cancer characterized by an unusual growth pattern with clusters of cells presenting abnormal polarity. The genetic alterations underlying this particular tumor type are still mostly unknown.

Whilst the dataset that the author collected is quite remarkable, especially the sequencing of several tumor cell clusters from matched primary tumors and lymph nodes metastasis, there's a lack of depth in the analyses, which reduces the impact of the findings. Copy number data is the main focus of the study and even though a selection of genes is identified, that may be associated with metastatic potential, further validation is necessary. While functional validation of the candidate role in favoring metastatic potential is probably beyond the scope for this study, the authors should at least consider analyzing publicly available datasets to confirm the potential for the selected genes to drive metastasis, or at a minimum, discuss their findings within the context of previous publications that have explored CNV profiles linked to IMPC (i.e. Natrajan *et al.* 2014, Marchio' *et al.* 2008, etc.). The authors conclude that their study 'fundamentally explains the genomic characteristics and evolutionary process of IMPC clustered growth, proliferation, invasion and metastasis'. While the data they generated as part of the study certainly has the potential to explore these essential questions, the presented manuscript does not appear to have conclusively resolved any of the above.

We very much appreciate the insightful comments. We have provided the following explanations accordingly.

The landscape of our manuscript is as follows. We first analyzed the frozen bulk IMPC samples (primary tumors and paired normal tissues, without metastases) to obtain the SNV and CNV data. During the analysis we found that CNV was more likely to be associated with high metastatic potential of IMPC than SNV. Therefore, we focused on the genomic CNV, which may play a key role in the high metastatic potential of IMPC. This is the reason why we further analyzed more IMPC FFPE samples (primary tumors and matched metastasis) to study the genomic CNV of IMPC. As the CNV analysis did not need a high sequencing depth, we performed low-depth WGS, which was not enough to analyze SNV. Therefore, the SNV data cannot further validate our CNV conclusion, such as the metastatic path we revealed by CNV.

Copy number data revealed some metastatic driver genes, such as *IGSF9*, *PRDM16* and *ALDH2*, and we validated these gene in TCGA dataset (for details, see the answer to question 5), and further validated gene expression in 86 IMPC tumor samples; detailed data are provided in the last paragraph of the results (**Figure 7**). We demonstrated that these three genes are correlated with IMPC metastatic potential and may serve as potential targets for IMPC diagnosis and therapy in the future.

Finally, as suggested, perhaps some conclusions in our manuscript were not appropriate, such as “fundamentally explains the genomic characteristics and evolutionary process of IMPC clustered growth, proliferation, invasion and metastasis”; therefore, we modified the conclusion in the revised manuscript.

General comments

1) One major criticism is that WES data is analyzed very superficially and immediately discarded as not relevant to tumor invasion and metastasis on a disputable basis. A better description of the mutational patterns of the studied tumors is necessary to conclude whether point mutations do really play a minor role. Also, SNV data would certainly help corroborate the evolutionary paths described in the last result paragraph.

We concur. As suggested, we updated the in-depth WES data analysis of somatic mutation signatures (excluding synonymous SNV, including SNV-associated pathway analysis, and comparing the findings with TCGA data) and tumor mutation burden (TMB) (**Figure 1** and **Figure S1**,). Details are provided in the revised manuscript in paragraph one of the results.

From the lens of mutation spectrum (**Figure R2-1**), except for *TP53*, *CDC27*, *CELSR2*, *CEP192*, *KIF26A*, *RYR2*, *TAF4* and *TPBGL*, the proportions of the other genes in the 17 samples was very small, generally less than 2; therefore, the significance of comparing the frequency of single genes appears limited.

Pathway analysis of all non-silent somatic genes revealed that the RTK-RAS pathway involves nearly half of the samples (52.94%, 9/17) (**Figure R2-2**), and the samples with more lymph node metastases were mostly enriched in this pathway. It indicates that the RTK-RAS pathway may be an important for IMPC metastasis/development (**Figure R2-3**).

Figure R2-1: None silent somatic mutation patterns and frequencies for the most mutated genes. Updated as **Figure 1A**.

Figure R2-2: IMPC bulk frozen samples with none silent somatic mutation and enrichment of oncogenic signaling pathways. This figure is also provided as **Figure S2C**.

Figure R2-3: The RTK-RAS pathway affects the most samples (9/17). The 10 genes (Y-axis) represent the number of gene mutations involved in the RTK-RAS pathway in our sample. Updated as **Figure S2D**.

Following the advice of the reviewers, we conducted a detailed correlation analysis to prove that SNVs do play a minor role in the metastatic potential IMPC. The analysis is as follows:

Based on the correlation analysis, the Jaccard similarity index (JSI) was used to quantify mutational similarity of the tumor metastatic potential at the SNV and CNV level and perform statistical tests. The results confirmed that the JSI values of CNV were significantly higher than those of SNV (student's *t*-test, P-value = 0.0003) (**Figure R2-4**).

Figure R2-4: Jaccard similarity index (JSI) between CNVs and SNVs. Updated as the **Figure S3**.

Furthermore, previous research has reported that CNVs play a major role in breast cancer, while SNVs play only a minor role. (Ciriello, G., Miller, M., Aksoy, B. *et al.* Emerging landscape of oncogenic signatures across human cancers. *Nat Genet* 45, 1127–1133 (2013). <https://doi.org/10.1038/ng.2762>)

Figure R2-5: Distribution of mutations and copy number changes in BRCA tumor type. Samples vary in the number of recurrent copy number alterations (X-axis) and the number of recurrent mutations (Y-axis) (Figures from Ciriello, *et al.* Nat Genet. 2013. Supplementary Figure 3).

Furthermore, we agree with the reviewer that it would be meaningful to include SNV data in the evolutionary analysis; however, due to the limitations in samples and data types, frozen tissue data were subjected to WES but without lymph node metastasis data, while cell clusters data were only subjected to low-depth WGS, and SNV analysis was limited. Therefore, we only analyzed the driver gene mutations in SNV data, and we found mutations in *IGSF9* and *PRDM16* but not *ALDH2*. Furthermore, *MUC1*, *MAPK1* and *ARHGEF10*, which are the driver genes of clonal evolution in a single sample (P23), were not detected in SNVs (**Supplementary Table 2**).

2) In several instances the conclusions reported at the end of a result paragraph are not a clear summary of the analysis reported in the same paragraph. As an example, paragraph 5 concludes that ‘the ability to develop a considerable amount of genetic variations in IMPC cells is a driving force critical for its high metastatic potential’, whereas the entire section was devoted to the comparison between special and genetic distance. This lack of clarity and linearity in the way data and conclusions are presented requires a lot of effort in trying to understand what the message is and if it is effectively supported by the data.

We thank the reviewer for this helpful comment. We concur that the conclusion of paragraph 5 was not clearly expressed. We have deleted it from the manuscript and provided a new conclusion. We also modified several sentences in the conclusion to make them more appropriate and more closely connected to the conclusion, such as the conclusion reported at the end of paragraph 3 of the results section.

3) Similarly, in several instances it is not immediately clear what a specific analysis is supposed to test or prove. Each analysis should be introduced with a clear hypothesis.

For example, in the second result paragraph it is not obvious what the Pyclone method is supposed to highlight. The conclusion that a modest difference in correlation coefficients is indicative of a more prominent role for CNV vs. SNV in determining the potential for lymph-node metastasis is not convincing. Having a clear statement of the expectations, a more detailed explanation of the methodology (i.e. which features were correlated?) and an interpretation of the results would help clarify the finding

Per the reviewer's suggestions, in order to have a clear statement of the results analysis, we have revised the second paragraph of the results section.

The Pyclone method is a statistical model for inference of clonal population structures in cancers. We have added a reference to the revised manuscript (Roth A, *et al.* PyClone: statistical inference of clonal population structure in cancer. *Nature methods* 11, 396-398 (2014)). In the second result paragraph of the results section, we have explained why we used the Pyclone method.

Moreover, to make our results more convincing, we employed two more analyses to confirm that the lymph node metastatic potential was more associated with CNV than SNV, as follows:

Based on the correlation analysis, the Jaccard similarity index (JSI) was used to quantify the mutational similarity of the tumor metastatic potential at the SNVs and CNVs level and perform statistical tests. The results confirmed that the JSI values of CNVs was significantly higher than those of SNVs (student's *t*-test, P-value = 0.0003) (**Figure R2-4** above).

Furthermore, previous research has reported that for breast cancer, CNVs play a major role, while SNVs play only a minor role (**Figure R2-5** above). (Ciriello, G., Miller, M., Aksoy, B. *et al.* Emerging landscape of oncogenic signatures across human cancers. *Nat Genet* 45, 1127–1133 (2013). <https://doi.org/10.1038/ng.2762>)

4) Overall, there's lack of clarity when reporting p-values: in most cases the statistical test used is not reported, and in several occasions, it is not clear what exactly is being compared (i.e. result paragraph 3: 'the CNV characteristics are significantly distinct between IMPC and IDC (P=0.037)'). P-values are mentioned in the text but not in the corresponding figures.

We appreciate the reviewer's suggestion. The comparison groups are now more clearly stated.

We also have reported the statistical test before each P-value in our revised manuscript. P-values are mentioned in the text and in the corresponding figures or legends.

We have revised the unclear description of P-values, such as in paragraph 3 of the results section, as follows: “These findings indicate that, while IMPC and IDC share substantial amounts of CNVs, the overall characteristics of CNVs are distinctly different between IMPC and IDC, including in the primary tumor and metastases (student’s *t*-test, $P=0.037$).” Furthermore, the comparison of IMPC and IDC was from a comprehensive level, and there were no corresponding figures requiring P-values; we have mentioned this point in the revised manuscript.

5) Ideally, all of the CNV data indicating a number of affected genes that are listed as candidate drivers of metastasis and IMPC phenotype should be supported by expression data in the presented dataset or, if this is not feasible due to lack of biological material, across independent published datasets.

We concur with the reviewer’s comment. The CNV data revealed three driver genes of metastasis, namely, *IGSF9*, *PRDM16* and *ALDH2*. We searched the independent published TCGA datasets, and found that they were indeed related to cancers. Immunohistochemical differences were also confirmed, and detailed information is provided in the last paragraph of the results section (**Figure 7**). The following is the results of *IGSF9*, *PRDM16* and *ALDH2* in the published TCGA dataset:

Figure R2-6: Low specificity of *IGSF9* RNA expression in human different cancers. *IGSF9*: The average FPKM of *IGSF9* in all samples is 9.0.

Figure R2-7: IGSF9 is not a prognostic factor in 1075 breast cancer patients from TCGA database. The results of IGSF9 come from <https://www.proteinatlas.org/ENSG00000085552-IGSF9/pathology/breast+cancer>.

FigureR2-8: Low specificity of PRDM16 RNA expression in human different cancers. PRDM16: The average FPKM of PRDM16 in all samples is 0.1.

Figure R2-9: PRDM16 is not a prognostic factor in 1075 breast cancer patients from TCGA database. PRDM16 inhibits the metastasis in lung adenocarcinoma (Fei LR, Huang WJ, Wang Y, *et al.* PRDM16 functions as a suppressor of lung adenocarcinoma metastasis. *J Exp Clin Cancer Res.* 2019; 38(1)), consistent with our conclusion that PRDM16 loss in IMPC will promote metastasis.

The results of PRDM16 come from <https://www.proteinatlas.org/ENSG00000142611-PRDM16/pathology/breast+cancer>.

FigureR2-10: Low specificity of ALDH2 RNA expression in human different cancers. ALDH2: The average FPKM of ALDH2 in all samples is 9.8.

Figure R2-11: ALDH2 is not a prognostic marker in 1075 breast cancer patients from TCGA database, but ALDH2 plays a different or even opposite roles in different cancers. For instance, in urothelial cancer high expression of ALDH2 predicts a poor prognosis, while in liver cancer, renal cancer and endometrial cancer, high expression of ALDH2 predicts a good prognosis.

The results of ALDH2 come from <https://www.proteinatlas.org/ENSG00000111275-ALDH2/pathology/breast+cancer>.

More detailed comments.

1. Since IMPC is not a well-known tumor type, a better introduction to the concept of tumor cell clusters would help the reader understand the peculiarity of this tumor type.

We have added a description of IMPC tumor cell clusters in introduction and provided the hematoxylin-eosin (H&E) staining and the epithelial membrane antigen (EMA) staining results in **Figure S1A** and **S1B**.

2. An introductory figure that describes the entire dataset would be very helpful as it is difficult to follow sample availability and sequencing methodology through the different result paragraphs. Similarly, a final schematic representation of the proposed metastatic path would help clarify the main message of the study.

We greatly appreciate your suggestions. Our entire dataset includes two parts, and the experimental flow chart is shown in **Figure S1C-D**. One part is the bulk frozen tissues from 17 paired IMPC samples in results 1-2, and the other is cell clusters data from 29 FFPE samples in results 3-5.

Furthermore, we have provided the revised **Supplementary Table 1** to describe the sample availability and sequencing methodology of the 17 paired bulk frozen tissues, and revised **Supplementary Table 5** to describe the sequencing methodology and sample availability of the 29 FFPE samples.

As suggested, we have also provided a final schematic representation of the monoclonal metastatic path, which is shown in updated **Figure S5C**.

3. Figure 1B. Simply stating the frequency of single nucleotide substitutions is very limited, especially given the recent flourish of tools to compute mutational signatures that consider the three-nucleotide context of each mutation.

We thank the reviewer for the suggestions. We added an analysis of 96 signatures based on three-nucleotide context and compared these with the TCGA data. The results show that C>T mutations account for the majority, which is consistent with our previous conclusion.

Figure R2-12: The left figure is the mutations of IMPC, and the right figure is the mutations of TCGA breast cancer.

4. Figure 1E: although the text references the TCGA BRCA dataset, including a statistical analysis with p.value, the figure does not report TCGA data at all, which is confusing. Also, a robust comparison between the authors' dataset and the TCGA BRCA dataset should at least incorporate ER, PR and HER2 status and make sure that mutation frequencies are only compared between matching sets of samples.

As suggested, we have provided TCGA data in the figures, including breast cancer and luminal B subtype breast cancer (**Figure 1B** and **1C**). Based on the previous research findings (Marchiò *et al.*), the mutation landscape of IMPC is similar to that of luminal B breast cancer. At the same time, 76.5% (13/17) of our frozen tissue samples were the luminal B subtype (**Supplementary Table 1**), thus, the results of all breast cancer and luminal B subtypes in TCGA were compared separately to avoid the bias caused by genomic differences between different subtypes of breast cancer. TCGA breast cancer data were downloaded from the Broad Institute TCGA Genome Data Analysis Center (2016): Mutation Analysis MutSig 2CV v3.1. TCGA luminal B subtypes breast cancer data downloaded from the published article. (<https://www.ncbi.nlm.nih.gov/pmc/articles/PMC5959730/> A comprehensive Pan-Cancer molecular study of gynecologic and breast cancers). We also clarified TCGA information in the methods.

5. Figure 2B. the text refers to a p.value which is not reported in the figure. No stats support. The legend is not comprehensive, for example it is not clear what each row represents (is it a genomic window of a specific size?), nor what the blue/red color scheme represents.

We thank the reviewer for this comment. The P-value is reported in the figure (**Figure 2E**), and the statistical analysis has been added before the P-value in the revised manuscript. The CNV map (**Figure 2A**) represents the union of CNV segments of 17 samples; therefore, each row is the common segment of the smallest sample of CNV. Because it is a union, the smallest segment is large or small and does not follow the size of the chromosome to scale. The color of the segment represents the CNV log ratio value ($\log_2 X - 1$), therefore, white indicates that CNV has not occurred, blue indicates a negative value, which represents loss, and red indicates a positive value, which represents gain (**Figure 2A**).

6. In the second result paragraph, the authors establish a classification of IMPCs into two distinct groups, based on their CNV profile. However, these groups are not further explored and none of the subsequent analyses refer back to this classification.

As suggested, we further analyzed the focal area genes with GISTIC2.0 (**Supplementary Table 4**), which contains different genes in the chromosome arm as well as cancer-related genes, such as the gain of *ERBB2* and loss of *PRDM16* and *STK11*. It is worth mentioning that the loss *PRDM16* showed significant differences in the CNV high and low groups (student's *t*-test, $P = 0.011$) (**Table R2-13**), which is

consistent with the result that *PRDM16* plays an important role in cell clusters data.

Groups	Patient ID	Chr1:35002:30495000 (copy number log ratio)	P-value (student's t -test)
CNV high group	P45	-0.41366	0.011
	P8	-0.14039	
	P22	-0.77048	
	P41	-0.33733	
	P37	0.0076392	
	P40	-0.47014	
	P35	0.096816	
	P14	-0.36397	
CNV low group	P11	-0.078411	
	P36	0.0558	
	P17	0.097397	
	P43	0.12314	
	P42	0.12977	
	P19	0.049114	
	P38	-0.0041546	
	P44	-0.049154	
	P39	-0.29627	

Table R2-13: The CNV of *PRDM16* is significantly different between CNV high group and low group (student's *t*-test, P-value = 0.011).

7. Figure 3. The entire paragraph 'CNVs in pure IMPC and mixed IMPC-IDC based on multicell-clusters sequencing' is hard to follow and its corresponding Figure 3 is difficult to interpret. According to the text, the samples represented in 3A should be pairs of IDC and IMPC, this is not highlighted in the figure. Also, this entire paragraph is based on intersecting genes affected by gains or losses in a very limited number of samples. It is hard to believe that these findings have statistical relevance and could be reproduced in a larger dataset. It would certainly be more compelling to present a matched analysis of the IMPC-IDC pairs from the same patient, which instead is missing.

We appreciate this advice. In order to easily follow, we revised the entire paragraph. We also revised the sample ordering and label symbols in **Figure 3A** to highlight the paired IMPC and IDC samples. In addition, we have added q values (False Discovery Rate) to define the significant alterations; the details are shown in revised **Supplementary Table 6**.

8. The use of amplification/deletion vs. gain/loss is quite confusing and should be clarified and kept consistent.

We thank the reviewer for this comment. In the revised manuscript, we checked the whole manuscript and unified it using gain/loss to describe the degree of CNV.

9. Figure 6. The figure must be annotated more clearly, with labels indicating the sample names in each subset.

Thank you for your suggestion. We have annotated these labels clearly in the updated **Figure 6**.

10. In the discussion, the authors mention immuno-histochemistry data confirming changes in the protein levels of three candidate genes (IGSF9, PRDM16 and ALDH2), however the data is not presented in any figures and it is unclear what exactly has been done.

We have added the immunohistochemistry results of the three key metastasis genes to the last paragraph of the results, as shown in red (**Figure 7**). We validated these three genes in tumor samples from 86 IMPC patients. Furthermore, correlations between the gene expression level and clinicopathology features are provided in **Supplementary Table 9**. The overall survival and disease-free survival of IMPC patients in the high gene expression group and low gene expression group are provided in **Figure 7G**, and the prognostic factors for overall survival and disease-free survival have been added to **Supplementary Table 10**.

Reviewer #3

This is an interesting study from Shi and colleagues, examining in detail the repertoire of mutations and copy number alterations in a series of invasive micropapillary carcinoma of the breast. However, the authors should elaborate their results more precise and concise, and prove substantial evidence in order to make solid conclusions.

We truly appreciate these favorable comments.

Major points are listed below,

1. The results regarding to the somatic mutations are rather unclear. Please provide a list of somatic mutations identified in primary IMPC and lymph-node metastasis, which should include mutation type, alternate allele, mutation allele fraction, and pathogenicity status.

Per the reviewer's insightful advice, we have provided partial nonsynonymous mutation genes of IMPC bulk samples in the updated **Figure 1**, and the complete nonsynonymous mutation genes are provided in the updated **Supplementary Table 2**. Metastatic lymph nodes were not analyzed for somatic mutations because of the low sequencing depth, but genomic CNV analysis of IMPC cell clusters was performed. The details of gene CNVs are provided in the third sheet of the **Supplementary Table 6**.

2. Only non-synonymous mutations should be considered to contribute to pathogenesis, the top 30 mutant genes should be reanalyzed.

Thank you for your suggestions. In the revised manuscript, we removed the synonymous mutations, and the top mutant genes were also reanalyzed; details are provided in the updated **Figure 1** and **Figure S2A**.

3. The frequencies of TP53 and RYR2 mutations in breast cancer in TCGA are not mentioned in the text, thus it is confusing that the authors conclude that only three genes as common mutant genes in IMPC.

We thank the reviewer for pointing out this shortcoming and updated the **Figure 1**. The IMPC mutation frequencies of *TP53*, *RYR2* and *PIK3CA* were 24%, 18% and 12%, respectively. We have added the mutation frequency into the revised text. Because we have reanalyzed the data, the previous conclusion has changed, and details are provided in the first paragraph of the results section.

4. The data of which version of TCGA the authors used to compare with those of IMPC. The authors should clarify this information in the methods. What histological types of breast cancer do the authors used for the comparisons?

We thank the reviewer for the valuable advice. We used Level 3 TCGA data to compare with IMPC. Approximately 76.5% (13/17) of our frozen tissue samples were the luminal B subtype (updated **Supplementary Table 1**). Thus, we compared TCGA all-subtype-breast-cancer and TCGA luminal B subtypes breast cancer with IMPC separately to avoid the bias caused by genomic differences between different subtypes of breast cancer (**Figure 1**). TCGA breast cancer data were downloaded from the Broad Institute TCGA Genome Data Analysis Center (2016): Mutation Analysis MutSig 2CV v3.1. TCGA luminal B subtypes breast cancer data downloaded from the published article. (<https://www.ncbi.nlm.nih.gov/pmc/articles/PMC5959730/> A comprehensive Pan-Cancer molecular study of gynecologic and breast cancers). We also clarified TCGA information in the methods.

5.Previous studies have shown that IMPC displays a mutation landscape similar to that of luminal B breast cancers. Further comparisons regarding mutations between IMPC and invasive breast cancers with the same immunohistological profile need to be done before the authors make the conclusion for the first paragraph of the result section.

We concur. According to previous research (Marchi`o *et al.*), the mutation landscape of IMPC is similar to that of luminal B breast cancer. Indeed, 76.5% (13/17) of our frozen tissue samples were of luminal B subtype (updated **Supplementary Table 1**). Therefore, TCGA all-subtype breast cancer and TCGA luminal B subtypes breast cancer were compared separately to avoid the bias caused by genomic differences between different subtypes of breast cancer (**Figure 1**).

6.For the CNV results in the second paragraph of the results, the authors write that “the most frequent recurrent changes 8p amplification” is incorrect. Previous studies reported a high frequency of 8q amplification in IMPC, which also appears to be the case in this work, as shown in Figure 2A. “8p, 17p and 22q” should be losses but not deletions.

Thank you for your thorough review. We apologize for the oversight. We have made corrections in the revised manuscript.

7.The authors claim two novel IMPC-specific genomic amplified regions in 10q14 and 10q22, what is the frequencies of these two regions in IMPC, and are they associated with any genes or clinicopathological features?

We appreciate the advice. In the revised manuscript, we performed a more detailed analysis of CNV. We found that the previously selected software (FACET) is more inclined to obtain large fragments of CNV regions, which is not conducive to our analysis at the gene level. Therefore, we employed the Control-FREEC analysis software (**Figure R3-1**). Although the results were slightly changed, the main conclusions remained unchanged, and relevant modifications have been made to the

revised manuscript (**Figure 2**).

Based on the new high-resolution CNV data, compared with the results of Marchi'o *et al.* in 2008, the new gained region is 10q22 (35.29%, 6/17) without 10q14. Further analysis revealed that there are 20 genes in the 10q22 region (**Supplementary Table 3**), one of which is a protein coding gene, *NUTM2B*, related to cancer. Studies have reported that the *NUTM2B* gene is associated with kidney clear cell sarcoma and endometrial stromal sarcoma. At present, there is no literatures reporting clinicopathological features related to 10q22 gains.

Figure R3-1: Comparison of the software before (FACET) and now (Control-FREEC). Control-FREEC can obtain more small fragments of CNV to analyze significant genes. IchorCNV was utilized to identify technical noise in the copy number profiles.

8.Statistic methods need to be written more clearly.

We thank the reviewer for the advice. We have made amendment accordingly.

9.Please rectify the color in the Figure 2B. Every one color should represent gain, loss, amplification and deletion, respectively.

Thanks. We have made amendments in **Figure 2B** (updated **Figure 2A**).

10.It is interesting that the authors elaborate the CNVs of IMPC, however, to conclude that CNVs are more significantly associated with lymph node metastasis rather than SNVs requires more comprehensive analyses.

We appreciate the comments. We have carried out more analysis to validate the potency of lymph node metastasis was more associated with CNV than SNV, as follows:

Based on the correlation analysis, the Jaccard similarity index (JSI) was used to quantify the mutational similarity of the tumor metastatic potential at the SNV and CNV

level and perform statistical tests. The results confirmed that the JSI values of CNVs were significantly higher than those of SNVs (student's *t*-test, P-value = 0.0003) (**Figure R3-2**).

Figure R3-2: Jaccard similarity index (JSI) between CNVs and SNVs. Updated as the **Figure S3**.

Furthermore, previous research reported that in breast cancer, CNVs play a major role, while SNVs play only a minor role. (Ciriello, G., Miller, M., Aksoy, B. *et al.* Emerging landscape of oncogenic signatures across human cancers. *Nat Genet* 45, 1127–1133 (2013). <https://doi.org/10.1038/ng.2762>)

Figure R3-3: Distribution of mutations and copy number changes in the BRCA tumor type. Samples vary in the number of recurrent copy number alterations (X-axis) and number of recurrent mutations (Y-axis) (Figures from Ciriello, *et al.* *Nat Genet.* 2013. Supplementary Figure 3).

Both methods indicated that the CNVs were more significantly associated with lymph node metastasis than SNVs.

11. The authors need to distinct gain and loss from amplification and deletion,

respectively, throughout the manuscript, including the figures and tables.

Thank you for the comment. In the revised manuscript, we checked the whole manuscript and unified the use of gain/loss to describe the degree of CNVs, including in the figures and tables.

12.For results in Supplementary Table 2, “common” means these genes share in all cases? Otherwise, frequencies need to be noted for each gene, as well as for those exclusively altered genes, and then define the recurrent exclusively altered genes in both pure-IMPC and IDC.

In the updated **Supplementary Table 6**, “common” means that these genes were shared in both comparison groups. To avoid ambiguity, we changed the terminology to “shared”. As suggested, we have noted the frequencies for each gene and sorted by frequency to annotate the recurrent exclusively altered genes in both comparison groups. The recurrent altered genes were present in at least 3 samples. The details are shown in revised **Supplementary Table 6**.

13.For the results in Supplementary Table 2. Base on the frequencies of gains and losses, statistical analyses should be done to define the significant alterations in each component, rather than simply find the exclusive altered genes.

We thank the reviewer for the comment. We added q values (False Discovery Rate) to define the significant alterations in each component; the details are provided in revised **Supplementary Table 6**.

14.For the third section of the results, I suppose the authors did the comparisons between all 8 pure IMPC and 21 mixed IMPC, as shown in the first sheet of Supplementary Table 2. If so, the gain of PRDM16, PRCC and ELF3 occurs in the IDC component from the eight mixed IMPC cases in the second sheet of Supplementary Table 2 should also appear in the first sheet in the exclusively in mixed-IMPC column. The number of cases that subjected for sequencing should be noted.

We thank the reviewer for the valuable comments. The first sheet of **Supplementary Table 6** is a comparison between 6 pure IMPC tumors and 17 IMPC components in mixed IMPC-IDC tumors, not the IDC component in mixed IMPC-IDC tumors. Therefore, the CNVs of *PRDM16*, *PRCC* and *ELF3* in the first sheet in the exclusively mixed IMPC-IDC column were losses. We have added an explanation to the revised manuscript.

Furthermore, as suggested, we summarized the numbers of cases that subjected for sequencing (**Table R3-4**), including the statistical methods and P-values. In addition, the number of cases that were subjected to sequencing have been added to the revised manuscript.

Positions	Histopathological classification	Pathological components for comparison	Cases	Statistical methods	P-values
Primary tumor	Pure IMPC	IMPC	6	Independent samples t -test	0.045
	Mixed IMPC-IDC	IMPC	17		
	Mixed IMPC-IDC	IMPC	8	Paired	0.028
	Mixed IMPC-IDC	IDC	8	Student's t -test	
Metastatic lymph node	Mixed IMPC-IDC	IMPC	6	Paired	0.033
	Mixed IMPC-IDC	IDC	6	Student's t -test	

Table R3-4: Summary of cases that were subjected for multiple cell-clusters sequencing for the third section of the results.

15.The authors identified deletions of TNFRSF14, MYC and PCM1 as truncal CNVs in primary IMPC and IDC. As far as I know, MYC is a proto-oncogene, while the authors find deletion of MYC as a truncal event in the primary IMPC and IDC really baffles me.

We thank the reviewer for the valuable comments. Regarding the properties of the *MYC* oncogene, we believe that its loss is not related to the occurrence of IMPC (this event is at least not a driver gene); thus, we removed this information from the manuscript and the corresponding **Figure 4**.

16.MUC1, MAPK1 and ARHGEF10 cannot be found in the first sheet of Supplementary Table 2, please clarify the type of alterations of these three genes.

We concur with the reviewer's concern. *MUC1*, *MAPK1* and *ARHGEF10* were the branch node genes discovered during the evolutionary relationship between IMPC and IDC components of P23 by single cell-cluster sequencing (**Supplementary Table 7**), which was an individual event and was different from **Supplementary Table 6**, which was presented the multiple cell-clusters information. The CNVs of *MUC1*, *MAPK1* and *ARHGEF10* in IMPC component of P23 were gain, loss and loss, respectively.

17.The functions of PRDM16 and IGSF9 are rarely investigated, especially in breast cancer, it would be interesting if the authors could engage some functional studies to prove the driven effects of these genes in IMPC metastasis.

We agree with the reviewer on the roles of *PRDM16* and *IGSF9* in IMPC metastasis. Although this is what we too wanted to do, at present, these functional studies for IMPC

appear not very closely related to the scope of this project and are difficult to perform. The detailed reasons may be the following:

The morphological features of IMPC include a cell cluster structure formed by the adhesion of multiple tumor cells to each other and a lack of a fibrovascular cores (**Figure R3-5**). Tumor cell clusters have the characteristics of polarity reversal and epithelial membrane antigen (EMA) expression on the outer edge of the tumor cell cluster cell membrane, which faces the interstitial side (**Figure R3-5**). IMPC is a good model for studying the invasion and metastatic properties of breast cancer, but the lack of a corresponding cell line and cell model makes it almost impossible to fully carry out basic research, which is a bottleneck that limits in-depth study of IMPC.

Figure R3-5: Morphological characteristics of IMPC tumor cell clusters with reversed polarity. The left figure shows HE staining and the right figure shows EMA staining of IMPC tumor cells.

In our research group, the long-term goal is to establish an IMPC cell line, and we have tried various methods, such as the cultivation of IMPC primary cells. We grew IMPC tumor cells in a three-dimensional culture gel droplet that can mimics the environment of the human environment. IMPC tumor cells successfully formed cell cluster structures in the gel droplets (**Figure R3-6**), but the morphology of the cell cluster could not be maintained through passaging. Therefore, it not feasible to use IMPC cell line to study the functions of key transfer genes.

Figure R3-6: Cultivation of IMPC primary tumor cells with three-dimensional culture gel droplets.

Additionally, our research group also constructed a patient-derived xenograft (PDX)-model, which refers to transplantation of fresh tumor tissue from postoperative/biopsy IMPC patients into immunodeficient mice, relying on the microenvironment provided by the mice to maintain tumor growth. Over 2 years, we included more than 10 cases of IMPC patients, and only 1 case successfully developed into a tumor. However, we subsequently found that the tumor cells were not typical IMPC tumor cell clusters, as verified by the IMPC-specific marker protein EMA (**Figure R3-7**).

Although in past years, we tried to establish cultured cell lines or mouse-tumor models to mimic IMPC, currently, there is no such model established. Thus, we could not perform functional validation experiments for these two genes.

Figure R3-7: IMPC patient-derived xenografts (PDX)-model. The left is a tumor developed from an immunodeficient mouse, and the right is the EMA staining of the tumor cells without the special polarity reversal growth pattern.

18.It is difficult for me to understand why *IGSF9* and *ALDH2* gene CNVs were not shown in the results of Figure 3 and Figure 4, due to the sequencing depths? As these gene CNVs are common features related to high metastatic potential as the authors claim.

We thank the reviewer for the suggestion. Part 3 of the **Figure 3** shows the results of the multiple cell-clusters sequencing of different paraffin samples, while the *IGSF9* and *ALDH2* genes were found to be the driver genes of progression from the primary tumor to metastases in the same patient by single cell-cluster analysis. The purpose of these two parts is different.

However, in part 3 of **Figure 3**, we certainly detected the loss of *IGSF9* and the gain of *ALDH2*. Moreover, *IGSF9* is one of the specific losses in IMPC, while the genes in the Venn diagram are not all genes with CNV; for details, please see the updated **Supplementary Table 6**.

Part 4 of **Figure 4** shows the evolutionary process between IDC and IMPC components in the same patient, P23. The genes presented in **Figure 4** are cancer-related genes associated with the branch node from IDC to IMPC. The loss of *IGSF9* and gain of *ALDH2* were certainly detected in P23. However, these two genes are not ranked highly in the branch node and thus are not present in the Figure; for details, please see the updated **Supplementary Table 7**.

19.The authors mention in the Discussion section that “The immunohistochemistry results of the three key metastatic genes, i.e., *IGSF9*, *PRDM16* and *ALDH2*, in clinical samples of IMPC and IDC have also been verified at the protein level.”, where are these results? If there are, correlations between each gene expression and clinicopathological features and prognosis should be calculated.

Per the suggestion, we have added the immunohistochemistry results for the three key metastasis genes to the last paragraph of the results (**Figure 7**) in red text. We validated these three genes in tumor samples of 86 IMPC patients. Furthermore, the correlations between gene expression and clinicopathology features are provided in **Supplementary Table 9**. The overall survival and disease-free survival of IMPC patients in the high gene expression group and low gene expression group were provided are provided in **Figure 7G**, and the prognostic factors for overall survival and disease-free survival were added to **Supplementary Table 10**.

20.Supplementary Table 1 and 4 should be merged, whist, there are 30 but not 29 cases in supplementary Table 4.

As suggested, we merged the **Supplementary Table S1 and S4**, trimmed some redundant information, such as “Gender” and “Age” and created a new updated **Supplementary Table 5**. Furthermore, we thank the reviewer for pointing out this error. As P5 failed to pass the DNA sample quality control, 29 IMPC samples were included in present study. However, in the original **Supplementary Table 4**, we enrolled P5 in the IMPC samples and obtained 30 IMPC patients. We apologize for our mistake. In the revised **Supplementary Table 5**, we have corrected the text.

21. The authors applied single cell-cluster sequencing data in both primary tumors and metastatic lymph nodes of six cases, and identified the deletion of IGSF9 and PRDM16 and amplification of ALDH2 as drivers for lymph node metastasis in IMPC. Among the six cases, one has both IMPC and IDC component metastases in lymph node that was sequenced. To also show that IGSF9 and ALDH2 are exclusively altered in the IMPC component will solid authors' conclusion.

These comments are truly appreciated. We have further checked the CNV data of P23 patient (**Supplementary Table 7**) and found that the genomic region of PRDM16 was gained in the IDC component, while *IGSF9* and *ALDH2* were not changed in the IDC component. Therefore, we can conclude that the loss of *IGSF9* and *PRDM16* and gain of *ALDH2* are exclusively altered in the IMPC component.

22. The results need to be reorganized and discussed accordingly and concisely in order to make it easier to follow.

Per the suggestion, we revised the discussion accordingly.

REVIEWER COMMENTS

Reviewer #1 (Remarks to the Author):

The authors have addressed my comments adequately

Reviewer #2 (Remarks to the Author):

The reviewed m/s has been improved both conceptually and stylistically. However, it would benefit from another round of editing, to improve the overall language, especially for the new text that has been introduced after the first round of revisions.

Below are some examples of sentences that require clarification, as well as some additional minor comments on content.

- Abstract. The sentence 'Formalin-fixed paraffin embedded (FFPE) 422 tumor cell clusters of primary IMPC and paired axillary lymph node metastases were also examined using WGS', should read: '422 formalin-fixed paraffin embedded (FFPE) tumor cell clusters of primary IMPC and paired axillary lymph node metastases were also examined using WGS'
- Result chapter 1.
- This sentence is needs to be clarified: Because of the small tissue size, the number of mutations in the whole sample was so limited that the frequency of mutations might not reflect the true condition of the sample; therefore, we show only the genes reported to have a high mutation frequency in breast cancer, such as PIK3CA, AKT1, MAP3K1, FOXA1, GATA3, NCOR1, PTEN, PTPRD, and RUNX1
- 'We used different numbers of characteristic gene mutations (N = 1 to 8) to analyze the heterogeneity of the samples'. This sentence needs clarification.
- Figure legend 1. 'none silent' should probably read 'non-silent' or 'non-synonymous'.
- I appreciate the authors' effort to use consistent terminology when referring to copy number loss and gain. However, without the accompanying term 'copy number', 'loss' is a very generic term and, without proper context, it rather suggests homozygous loss of function. The authors should be more precise, especially in the abstract (i.e. The PRDM16, ELF3 and PRCC genes were lost in primary IMPC tumor cell clusters...).
- There are instances where the authors addressed the reviewer's comments in the rebuttal but did not made the appropriate adjustments/edits to the m/s. For example, the reviewer's request to analyze mutational signatures (and mutations in their trinucleotide context) has been discussed in the rebuttal but no mention of this analysis has been added to the m/s. I still believe that an analysis of the WES data using tools such as deconstructSigs (as an example) would add value to the analysis, by identifying mutational signatures present in the tumor samples without the need to develop de novo signatures, which may not be feasible given the modest size of the dataset. Where it is correct

that C>T mutations are among the most common in cancer, understanding the context in which they arise would allow to discriminate between those generated in aging cells, those that are linked to APOBEC activity, those that are due to defects in DNA mismatch repair, etc...

Additional comments

The authors should add two columns to supplementary table S2 reporting the variant allele frequency of each reported somatic mutation. Ideally an estimate of pathogenicity (e.g. one of 'neutral', 'low', 'medium' and 'high' as predicted by MutationAssessor, or by alternative publicly available tools)

In the copy number analysis result paragraph, the authors should report the frequency of the copy number changes across their dataset. For example, they identify chr10q22.3 as a novel region of amplification specific to IMPC, but there is no indication of how prevalent this genomic alteration is.

The authors' response to comment no.10 is not satisfactory: The authors' choice to focus on CNVs, rather than SNVs, in their analysis of genomic determinants of metastatic potential still does not appear to be strongly substantiated. The authors introduce the use of the Jaccard similarity index as an alternative way to determine the association between genomic features and metastatic potential, however the rationale of this analysis is not clear. Since this is not a commonly used method to compare the mutational burdens associated with CNV vs. SNV, a better description of the rationale, methodology and results is required. In addition, the authors cite work from Ciriello et al. showing significant predominance of recurrent copy number alterations in breast cancer. It would be helpful to see the same trend reproduced in the authors' own tumor cohort, rather than simply using the published observation as evidence. In general, the entire result paragraph dedicated to this topic needs to be clarified.

The authors should consider defining gain, loss, amplification and deletion and use each term in the appropriate context. For example, while gain and amplification should not be used interchangeably, there is value in using the term 'amplification' rather than 'gain' in the specific context where the copy number gain is greater than a certain defined threshold.

The frequency of the shared gene CNVs in revised Table S6 should be shown for both tumor components so that a fair comparison of frequencies can be made (e.g. in the first sheet, 'pure-IMPC VS mix-IMPC', the shared frequency column should report the number of tumors with each reported alteration for pure-IMPC and mix-IMPC separately).

Tables S6 and S7 do not present results in an intuitive manner. Since they are presented as tables, each row should be read across all columns (which is not the case, currently). Also, for a more direct comparison, frequencies of gain/loss should be reported for all tumor cohorts. For example, the frequency of MUC1 loss in P-IMPC (Table S7) is 18/18, but no frequency is reported for P-IDC, which makes it hard to compare and interpret the data. Also, is MUC1 gained or lost? There is no clarity about this as the text reports that 'The gain of MUC1 gene and the loss of the MAPK1 and ARHGGEF10 genes play key roles in evolutionary branch nodes' (which, is not a clear statement), while the associated Table S7 only reports significant loss of MUC1 in both P-IMPC and M-IMPC, but no gain.

The results still need some reorganizing and clarification to improve the flow.

REVIEWER COMMENTS

The Reviewers' comments are underlined.

Reviewer #1 (Remarks to the Author):

The authors have addressed my comments adequately

Reviewer #2 (Remarks to the Author):

The reviewed m/s has been improved both conceptually and stylistically. However, it would benefit from another round of editing, to improve the overall language, especially for the new text that has been introduced after the first round of revisions.

We appreciate the comment from the reviewer. As suggested, we have revised the manuscript, particularly the new text that was inserted after the first round of revisions.

Below are some examples of sentences that require clarification, as well as some additional minor comments on content.

• Abstract. The sentence 'Formalin-fixed paraffin embedded (FFPE) 422 tumor cell clusters of primary IMPC and paired axillary lymph node metastases were also examined using WGS', should read: '422 formalin-fixed paraffin embedded (FFPE) tumor cell clusters of primary IMPC and paired axillary lymph node metastases were also examined using WGS'

We thank the reviewer for noting this problem. We have corrected the text in the revised manuscript.

• Result chapter 1.

• This sentence is needs to be clarified: Because of the small tissue size, the number of mutations in the whole sample was so limited that the frequency of mutations might not reflect the true condition of the sample; therefore, we show only the genes reported to have a high mutation frequency in breast cancer, such as PIK3CA, AKT1, MAP3K1, FOXA1, GATA3, NCOR1, PTEN, PTPRD, and RUNX1

We thank the reviewer for the suggestion. We have rewritten this sentence in the revised manuscript, as follows: "Additionally, the most frequently mutated genes in IMPC were *TP53* (24%, 4/17), followed by *CDC27*, *CELSR2*, *CEP192*, *KIF26A*, and *RYR2* (18%, 3/17, Fig. 1a), while the other mutated genes were present in fewer than 3 samples. Since the number of mutations in the 17 samples might be limited to such an extent that the frequency of gene mutations might not reflect the true genetic alteration spectrum of IMPC, we also examined the genes that were reported to have a high mutation frequency in breast cancer²⁴, such as *PIK3CA*, *AKT1*, *MAP3K1*, *FOXA1*, *GATA3*, *NCOR1*, *PTEN*, *PTPRD*, and *RUNX1* (Fig. 1a).".

• ‘We used different numbers of characteristic gene mutations (N = 1 to 8) to analyze the heterogeneity of the samples’. This sentence needs clarification.

We thank the reviewer for the comment. We have rewritten the sentence in the revised manuscript as follows: “In order to analyze the heterogeneity of the SNV data, a gradient threshold was set to determine the numbers of characteristic gene mutations (N). The correlation between mutational heterogeneity and the potential of lymph node metastasis peaked at N=5 and then gradually decreased, suggesting that an increase in the value of N would be meaningless. We therefore set the gradient threshold to 1-8 (Fig. 2f).”

• Figure legend 1. ‘none silent’ should probably read ‘non-silent’ or ‘non-synonymous’.

We appreciate the reviewer’s suggestion. We have changed “none silent” to “non-silent” in the revised manuscript including the Figure legend 1.

• I appreciate the authors’ effort to use consistent terminology when referring to copy number loss and gain. However, without the accompanying term ‘copy number’, ‘loss’ is a very generic term and, without proper context, it rather suggests homozygous loss of function. The authors should be more precise, especially in the abstract (i.e. The PRDM16, ELF3 and PRCC genes were lost in primary IMPC tumor cell clusters...).

According to the reviewer’s suggestion and to state the copy number status more precisely, we have changed the description of CNV states in the new version of revised manuscript including the sentence mentioned by the reviewer in the abstract.

• There are instances where the authors addressed the reviewer’s comments in the rebuttal but did not made the appropriate adjustments/edits to the m/s. For example, the reviewer’s request to analyze mutational signatures (and mutations in their trinucleotide context) has been discussed in the rebuttal but no mention of this analysis has been added to the m/s. I still believe that an analysis of the WES data using tools such as deconstructSigs (as an example) would add value to the analysis, by identifying mutational signatures present in the tumor samples without the need to develop de novo signatures, which may not be feasible given the modest size of the dataset. Where it is correct that C>T mutations are among the most common in cancer, understanding the context in which they arise would allow to discriminate between those generated in aging cells, those that are linked to APOBEC activity, those that are due to defects in DNA mismatch repair, etc...

We thank the reviewer for the suggestion. We have conducted an analysis of 96 signatures based on the three-nucleotide context and compared these signatures with TCGA data. C>T mutations account for the majority, which is consistent with TCGA breast cancer data. We have added these results to the new revised manuscript, as

updated in **Supplementary Fig. 3**.

In addition, we have also used deconstructSigs to analyze mutational signatures. However, we postulated that our sample size might be too limited to obtain conclusive results, and therefore did not include the data in the previous revision. The deconstructSigs results revealed that mutational signatures of APOBEC (signature 2 and 13), age (signature 1), homologous recombination deficiency (HRD) (signature 3) and DNA mismatch repair signatures (signature 6 and 15) were relatively higher in our IMPC samples (**Supplementary Fig. 4 and Supplementary Data 3**) (see below). We have now added the results to the new revision.

Supplementary Fig. 3: The analysis of 96 mutational signatures based on the three-nucleotide context in IMPC and TCGA breast cancer tissues. A The signatures of IMPC. **B** The signatures of TCGA breast cancer. Six categories of mutations (C > G, T > A, C > A, T > G, C > T, and T > C) are marked with different colors. The horizontal axis indicates different bases, and the vertical axis indicates the proportions of mutational categories.

Supplementary Fig. 4: The mutational signatures in 17 IMPC samples analyzed using deconstructSigs. Twelve mutational signatures were identified. The horizontal axis indicates different samples, and the vertical axis indicates different signatures. Notably, 0-0.6 represents the proportion of the distribution of mutational signatures in different samples.

Additional comments

The authors should add two columns to supplementary table S2 reporting the variant allele frequency of each reported somatic mutation. Ideally an estimate of pathogenicity (e.g. one of 'neutral', 'low', 'medium' and 'high' as predicted by MutationAssessor, or by alternative publicly available tools).

We appreciate the reviewer's comment. As suggested, we added the variant allele reads, variant allele frequency and related information to updated **Supplementary Data 2** (columns M-R). Meanwhile, an estimate of pathogenicity predicted by MutationAssessor was also added to the updated **Supplementary Data 2** (column S).

In the copy number analysis result paragraph, the authors should report the frequency of the copy number changes across their dataset. For example, they identify chr10q22.3

as a novel region of amplification specific to IMPC, but there is no indication of how prevalent this genomic alteration is.

We thank the reviewer for this comment. We have counted the frequency of the copy number gains and losses at the arm level in previous Table S3 (E and L column in sheet 1), now updated as **Supplementary Data 4**.

As suggested, we identified the copy number changes by following two rules at the arm level and CNV focal areas. One is defined as a gain/loss directly based on the changes in copy number (gain as a copy number ≥ 3 , loss as a copy number ≤ 1 and a copy number in between as normal), and the other is classified as deletion/loss/normal/gain/amplification based on the first classification according to the results of GISTIC2.0 and the recommended threshold calculated by the GISTIC2.0 algorithm (deletion ($\log \text{ ratio} < -1.3$), loss ($-1.3 \leq \log \text{ ratio} \leq -0.1$), normal ($-0.1 < \log \text{ ratio} < 0.1$), gain ($0.1 \leq \log \text{ ratio} \leq 0.9$) and amplification ($\log \text{ ratio} > 0.9$)). The details are provided in sheet 1 P column and sheet 3 B column (updated Supplementary Data 4). For instance, in the chr10q22.3 area, a gain was detected in 3 samples (17.65%, 3/17), and amplification was detected in 3 samples (17.65%, 3/17). Notably, although this region was not the highest frequency area, it is a region with significant copy number change ($q < 0.25$). This region has not been reported in previous studies; thus, we described it as a new CNV focal region in the results section.

The authors' response to comment no.10 is not satisfactory: The authors' choice to focus on CNVs, rather than SNVs, in their analysis of genomic determinants of metastatic potential still does not appear to be strongly substantiated. The authors introduce the use of the Jaccard similarity index as an alternative way to determine the association between genomic features and metastatic potential, however the rationale of this analysis is not clear. Since this is not a commonly used method to compare the mutational burdens associated with CNV vs. SNV, a better description of the rationale, methodology and results is required. In addition, the authors cite work from Ciriello et al. showing significant predominance of recurrent copy number alterations in breast cancer. It would be helpful to see the same trend reproduced in the authors' own tumor cohort, rather than simply using the published observation as evidence. In general, the entire result paragraph dedicated to this topic needs to be clarified.

We appreciate the comment from the reviewer. The jaccard similarity index (JSI) is an index used to quantify mutational similarities in different lesions. For example, recently, Zheng Hu *et al.* used the JSI method to analyze the prevalence of the polyclonal seeding between LNM and distant metastases (Hu Z, Li Z, Ma Z, Curtis C. Multi-cancer analysis of clonality and the timing of systemic spread in paired primary tumors and metastases. *Nature genetics* 52, 701-708 (2020)). In our study, we employed the same method to evaluate the mutational similarity of the tumor metastatic potential between the CNV level and SNV level, and found that the tumor metastatic potential was more prevalent at the CNV level.

Furthermore, according to the reviewer's suggestion, we referred to the method reported by Ciriello *et al.* to calculate the difference between SNV and CNV burdens using our own IMPC data set, and found that CNV burden was higher than the SNV burden (updated **Supplementary Fig. 5B**). The result was similar to the findings reported by Ciriello *et al.* in breast cancer, consistent with the hypothesis that CNVs plays a more important driving role in IMPC. We have added these statements in the text.

Supplementary Fig. 5B: Distribution of mutations (SNVs) and CNVs in the IMPC dataset. The 17 IMPC samples vary in the number of recurrent CNVs (X-axis) and number of recurrent SNVs (Y-axis). Gray represent one sample. Black represents two samples.

The authors should consider defining gain, loss, amplification and deletion and use each term in the appropriate context. For example, while gain and amplification should not be used interchangeably, there is value in using the term ‘amplification’ rather than

'gain' in the specific context where the copy number gain is greater than a certain defined threshold.

According to the reviewer's suggestion and based on the results of the GISTIC2.0 analysis and the recommended threshold calculated by the GISTIC2.0 algorithm, we divided the CNVs into five states: deletion ($\log \text{ratio} < -1.3$), loss ($-1.3 \cong \log \text{ratio} \cong -0.1$), normal ($-0.1 < \log \text{ratio} < 0.1$), gain ($0.1 \cong \log \text{ratio} \cong 0.9$) and amplification ($\log \text{ratio} > 0.9$). The threshold for gain and amplification is 0.9.

The frequency of the shared gene CNVs in revised Table S6 should be shown for both tumor components so that a fair comparison of frequencies can be made (e.g. in the first sheet, 'pure-IMPC VS mix-IMPC', the shared frequency column should report the number of tumors with each reported alteration for pure-IMPC and mix-IMPC separately).

As suggested, we added the frequency of the shared gene CNVs for both pure IMPC and mixed IMPC (H column in the first sheet). Sheets 2 and 3 contain data for paired samples, and thus the shared frequency column represents the number of tumors with both components (G column in sheets 2 and 3). Table S6 is now updated as **Supplementary Data 6**.

Tables S6 and S7 do not present results in an intuitive manner. Since they are presented as tables, each row should be read across all columns (which is not the case, currently). Also, for a more direct comparison, frequencies of gain/loss should be reported for all tumor cohorts. For example, the frequency of MUC1 loss in P-IMPC (Table S7) is 18/18, but no frequency is reported for P-IDC, which makes it hard to compare and interpret the data. Also, is MUC1 gained or lost? There is no clarity about this as the text reports that 'The gain of MUC1 gene and the loss of the MAPK1 and ARHGEF10 genes play key roles in evolutionary branch nodes' (which, is not a clear statement), while the associated Table S7 only reports significant loss of MUC1 in both P-IMPC and M-IMPC, but no gain.

These suggestions are good. In Table S6 (updated as **Supplementary Data 6**), our aim is to highlight the exclusive gene CNVs and shared gene CNVs for each tumor component. Therefore, the information presented in **Supplementary Data 6** may better explain the results shown in **Fig. 3b-d**. In addition, as suggested, we reformatted Table S7 (updated as **Supplementary Data 7**) to ensure that each row reads across each column, and we also reported the frequency of all tumor cohorts.

Additionally, we repeated the analysis of the CNV in MUC1, and found that MUC1 was indeed lost. We thank the reviewer very much for noting this error and made a correction in the new version of the revised manuscript.

The results still need some reorganizing and clarification to improve the flow.

According to the suggestion, we have tried to the best of our ability to reorganize and clarify the results presented in the revised manuscript.

REVIEWERS' COMMENTS

Reviewer #2 (Remarks to the Author):

The authors have addressed my comments adequately.

Note from the Editor

This manuscript was substantially corrected and re-evaluated by the Peer Reviewers during the Production stage. The Authors introduced the following modifications to the final version, which did not alter the main conclusions:

* The authors realised that the version of the cell cluster data used in their analysis was not the final filtered version. Some data with low quality had not been removed, and some data that passed their filters were absent. This influenced the CNV results and induced errors in the evolutionary tree, leading to errors in several figures.

* The authors corrected the CNV frequency in IMPC and IDC tumours and generated new fig. 3e to keep the same style and resolution as fig. 2b. In order to correct the CNV frequencies, they updated their data list, removed unqualified cell clusters and added qualified clusters, increased the total cell clusters from 422 to 442. They also found that the CNV caller ichorCNA introduced false positive CNVs, so they used the “DNA copy” package for all their CNV calling instead.

* They also corrected Fig. 6a. They checked all cell clusters and corrected the phylogenetic tree. The lymphatic metastasis was still inferred to originate from monoclonal metastatic seeds, but the evolutionary relationship within different lymph nodes was not clear. Therefore, they updated the information and rebuilt the phylogenetic tree of Figure 6a and Supplementary Figure 7. The legend, information about the clusters and their corresponding description were updated in the manuscript and Supplementary Data 5.

* To correct the cell clusters, the authors re-analysed the raw data and filtered clusters meeting the following criteria: $MAPD > 0.4$ or mapping rate $< 70\%$ or depth < 0.1 or coverage $< 1\%$. The initial evolutionary tree was based on inaccurate CNV data, so the cell cluster number and the evolutionary path were incorrect.

* When rebuilding the tree, the authors changed the method and used MEDALT (Wang, F. et al., Genome Biol 2021), which the authors claim is more accurate for inferring copy number lineages.

REVIEWER COMMENTS

Reviewer #1 (Remarks to the Author):

The latest manuscript represents a substantially improved version of the initial submission. The few methodological refinements did not alter the main conclusions of the paper and overall I support its publication.

Reviewer #2 (Remarks to the Author):

The authors have addressed my comments adequately.